# Shifts in Greenland interannual climate variability lead Dansgaard-Oeschger abrupt warming by hundreds of years

Chloe A. Brashear[1], Tyler R. Jones[1], Valerie Morris[1], Bruce H. Vaughn[1], William H.G. Roberts[2], William B. Skorski[1], Abigail G. Hughes[1], Richard Nunn[3], Sune Olander Rasmussen[4], Kurt M. Cuffey[5], Bo M. Vinther[4], Todd Sowers[6], Christo Buizert[7], Vasileios Gkinis[4], Christian Holme[4], Mari F. Jensen[8], Sofia E. Kjellman[10], Petra M. Langebroek[9], Florian Mekhaldi[11], Kevin S. Rozmiarek[1], Jonathan W. Rheinlænder[8], Margit H. Simon[9], Giulia Sinnl[4], Silje Smith-Johnsen[8], James W.C. White[12]

[1] Institute of Arctic and Alpine Research, University of Colorado, Boulder, CO, USA
[2] Geography and Environmental Sciences, Northumbria University, Newcastle-upon-Tyne, UK
[3] National Science Foundation Ice Core Facility, Lakewood, CO, USA
[4] Centre for Ice and Climate, Section for the Physics of Ice, Climate, and Earth, Niels Bohr Institute, University of Copenhagen, Copenhagen, Denmark
[5] Department of Geography, University of California, Berkeley, CA, USA
[6] Department of Geosciences, Pennsylvania State University, University Park, Pennsylvania, USA
[7] College of Earth, Ocean and Atmospheric Sciences, Oregon State University, Corvallis, Oregon, USA
[8] Bjerknes Center for Climate Research, University of Bergen, Bergen, Norway
[9] NORCE Norwegian Research Centre, Bjerknes Center for Climate Research, Bergen, Norway
[10] Department of Geosciences, UiT The Arctic University of Norway, Tromsø, Norway
[11] Department of Geology, Lund University, Lund, Sweden
[12] College of Arts and Sciences, University of North Carolina, Chapel Hill, NC, USA

*Correspondence to:* Chloe Brashear (chloe.brashear@colorado.edu)

**Abstract.** During the Last Glacial Period (LGP), Greenland experienced approximately thirty abrupt warming phases, known as Dansgaard-Oeschger (D-O) Events, followed by cooling back to baseline glacial conditions. Studies of mean climate change across warming transitions reveal indistinguishable phase-offsets between shifts in temperature, dust, sea salt, accumulation and moisture source, thus preventing a comprehensive understanding of the "anatomy" of D-O cycles (Capron et al,. 2021). One aspect of abrupt change that has not been systematically assessed is how high-frequency, interannual-scale climatic variability surrounding centennial-scale mean temperature changes across D-O transitions. Here, we utilize the EGRIP ice core high-resolution water isotope record, a proxy for temperature and atmospheric circulation, to quantify the amplitude of 7-15 year isotopic variability for D-O events 2-13, the Younger Dryas and the Bølling-Allerød. On average, cold stadial periods consistently exhibit greater variability than warm interstadial periods. Most notably, we often find that reductions in the amplitude of the 7-15 year band led abrupt D-O warmings by hundreds of years. Such a large phase offset between two climate parameters in a Greenland ice core has never been documented for D-O cycles. However, similar centennial lead times have been found in proxies of Norwegian Sea ice cover relative to abrupt Greenland warming (Sadatzki et al., 2020). Using HadCM3, a fully coupled general circulation model, we assess the effects of sea ice on 7-15 year temperature variability at EGRIP. For a range of stadial and interstadial conditions, we find a strong relationship in line with our observations between colder simulated mean temperature and enhanced temperature variability at the EGRIP location. We also find a robust correlation between year-to-year North Atlantic sea-ice fluctuations and the strength of interannual-scale temperature variability at EGRIP. Together, paleoclimate proxy evidence and model simulations suggest that sea ice plays a substantial role in high-frequency climate variability prior to D-O warming. This provides a clue about the anatomy of D-O Events and should be the target of future sea-ice model studies.

# 1 Introduction

1.1 Dansgaard-Oeschger Events

Stable isotopes of hydrogen ($\delta$D) and oxygen ($\delta^{18}$O) in polar ice cores provide information about local temperature and atmospheric circulation (Dansgaard, 1964). In Greenland, measurements of $\delta^{18}$O and $\delta$D during the Last Glacial Period (LGP; 115-12 ka) reveal recurrent alternations, referred to as Dansgaard-Oeschger (D-O) cycles, between quasi-stable warm periods (i.e. Greenland Interstadials; GI) and cold baseline conditions (i.e. Greenland Stadials; GS) (Dansgaard et al., 1993). Global climate change associated with GI phases includes increased aridity across North America and Eurasia (Wagner et al., 2010; Asmerom et al., 2010), a northward shift of the tropical rain belt (Cruz et al., 2005; Deplazes et al., 2013), and antiphase warm periods in Antarctica (Blunier et al., 1998; Blunier & Brook, 2001; EPICA Community Members, 2006). It is widely accepted that these changes are linked via fluctuating strengths of the Atlantic Meridional Ocean Circulation (AMOC), in which GI and GS phases are characterized by strong and weak overturning, respectively (Broecker, 1998; Knutti et al., 2004, Stocker & Johnsen 2003). Abrupt GS-GI transitions have received widespread attention due to the unusual rate and magnitude of climatic change. For example, Greenland temperature increases of 5-16.5 °C occurred in a matter of decades (Huber et al., 2006b; Kindler et al., 2014; Severinghaus et al., 1998; Severinghaus & Brook, 1999). Despite decades of research, the trigger of this phenomenon and associated global changes is still debated.

1.2 Sea Ice

North Atlantic sea ice is often referenced in explanations of D-O warming because of its rapid response to relatively weak forcing and its influence on both atmospheric and oceanic dynamics. Several model simulations (Li et al., 2005; Li et al., 2010; Boers et al., 2018; Dokken et al., 2013; Petersen et al. 2013) show sea-ice displacements in the North Atlantic and Nordic Seas, complimented by associated feedbacks, can produce abrupt temperature increases consistent with those documented by water isotopes in Greenland ice cores. Though model reconstructions vary slightly, a primary mechanism for abrupt warming includes initial sea-ice reductions amplified by oceanic heat loss to the atmosphere, perturbations to the ocean salinity gradient, and decreased surface albedo.

Determining the driver(s) of sea-ice displacements, in relation to abrupt Greenland warming, requires a highly resolved chronology of pertinent regional changes. Biomarker evidence from ocean sediment cores confirms that sea ice in the southern Norwegian Sea consistently experienced millennial-scale oscillations in pace with D-O cycles (Hoff et al., 2016). In one core location, ice cover reductions began during the late GS phase, thus leading abrupt warming by hundreds of years (Sadatzki et al., 2019, Sadatzki et al., 2020). However, these ocean sediment cores have low-resolution (i.e. multi-decadal scale sampling) and their age chronologies have large uncertainties. By comparison, Greenland ice cores can reach annual or multi-year resolution. For this reason, they have been key in establishing leads and lags of various geological parameters, deducing causal relationships, and disentangling the spatiotemporal progression of abrupt climate change. Observations of dust, moisture source and accumulation during D-O warming exhibit indistinguishable (Capron et al., 2021) or near synchronous (Steffensen et al., 2008) shifts, suggesting

widespread regional climate changes in the North Atlantic and Greenland occurred in phase. This is partly at odds with evidence from Norwegian Sea sediment cores.

1.3 Climate Variability

To further investigate leads and lags during abrupt D-O warmings, we utilize the EGRIP water isotope ($\delta$D) record to assess the temporal evolution of mean temperature vs. high-frequency water isotope variability in Northeastern Greenland. Variability in Greenland ice cores has long been studied, but coarse analytical sampling on increasingly compressed annual ice layers has limited LGP analyses to centennial- or millennial-scales (WAIS Divide Project Members, 2015; Rehfeld et al., 2018; Schulz et al., 2004). Similarly, interpretations of interannual- and decadal-scale Greenland climate variability have primarily been available for the Holocene (11.7 – 0 ka) (Hughes et al., 2020; Rimbu & Lohmann, 2010; Rimbu et al., 2021; Ortega et al., 2014). Though studies assessing decadal-scale variability during the LGP exist (Boers et al., 2018; Ditlevsen et al., 2002), the data sets used were discretely sampled at cm-scale resolutions which may diminish or conceal important high-frequency climatic information (Fig. A1). Developments in continuous-flow sampling techniques have recently allowed for high-frequency analysis of water isotope variability in Antarctic ice cores during the LGP by preserving the amplitude of interannual-scale signals (Jones et al., 2017a; Jones et al., 2018). In the case of West Antarctica, a shift in LGP interannual isotopic variability was linked to broad changes in Pacific Basin teleconnection strength driven by reductions in Laurentide Ice Sheet topography and changing albedo (Jones et al., 2018). This study demonstrated that the drivers of high-frequency climate variability can temporally decouple from the drivers of mean local climate (e.g. temperature, accumulation, etc.), providing new insights about paleoclimate dynamics.

In the northern high latitudes, sea ice varies substantially on multi-year and multi-decade bases, imparting variability into the climate system on similar timescales. The Greenland water isotope variability record may therefore provide clues about high-frequency sea-ice variations as such shifts would affect the isotopic signature of precipitation via influences on both moisture source and atmospheric circulation. Our study comprises of four parts: 1) comparison of LGP isotopic variability relative to the Holocene, 2) comparison of isotopic variability between warm Greenland Interstadials (GI) and cold Greenland Stadials (GS), 3) a moving window assessment of lead-lag relationships between isotopic variability and mean temperature increases associated with D-O Events and 4) a model assessment of LGP North Atlantic temperature and sea-ice variability at interannual scales using the general circulation model HadCM3, including how simulated sea-ice variability affects temperature variability at the EGRIP site.

**2 Methods**

2.1 Water Isotope Data

The EGRIP (East Greenland Ice Core Project) ice core was drilled at 75°38′ N and 35°60′ W from an interior location of the North East Greenland Ice Stream (NEGIS). Due to rapid ice flow in this location, glacial portions of the EGRIP ice core originate ~200 kilometers upstream near the Greenland Ice Sheet divide (Gerber et al., 2021). $\delta$D

and $\delta^{18}O$ records were measured using a semi-automated continuous flow analysis (CFA) system in conjunction with
a Picarro cavity ring down laser spectrometer (CRDS; model L2130-i) (Jones et al., 2017a). The CFA-CRDS system
is constituted by three subsystems: the ice core melter, a liquid-to-vapor phase conversion, and the isotopic analyzer.
Ice core sticks measuring $100 \times 1.3 \times 1.3$ cm are melted at a rate of 2.5 cm/min by an aluminum plate maintained at
14.6°C. Melt water is pulled through the preparation line via peristaltic pumps where it is filtered to remove
particulates, degassed, and aerosolized by a nebulizer. Finally, ice core derived water vapor is introduced to the
CRDS in a continuous, stable stream for optimal performance.  Precise $\delta D$ and $\delta^{18}O$ measurements are derived in real
time relative to Vienna Standard Mean Ocean Water (VSMOW; $\delta^{18}O = \delta D = 0$ per mil). Using this CFA
methodology, the water isotope record has a sample resolution of ~1mm throughout, with an effective resolution of
~5 mm due to mixing. Figure A1 demonstrates the artificial loss of interannual amplitudes with depth for coarser (i.e.
cm-scale) sampling resolutions.

To date, the EGRIP water isotope record extends to ~49.9 ka b2k (thousands of years before 2000 CE) according to
the Greenland Ice Core Chronology (GICC05) (Svensson et al., 2008) (Fig. 1a). The GICC05 was transferred to
EGRIP using 373 tie points between 0-14.96 ka b2k (Mojtabavi et al., 2019) and an additional 138 tie points between
14.96-49.9 ka b2k (Gerber et al., 2021). For each EGRIP data point (in mm-resolution), the corresponding NGRIP
depth was obtained by linear interpolation between tie points and age was determined from the annual-layer counted
GICC05 timescale. The ages used here inherit the maximum counting error of the GICC05 timescale. In addition, the
interpolation introduces an uncertainty which is relevant when comparing to other records on the GICC05 timescale.
Isotope data is interpolated at a uniform time interval of 0.05 years. On average, temporal differences in adjacent data
points (i.e. next to one another) range from sub-weekly in the Holocene to sub-monthly during the LGP.

## 2.2 Spectral Analysis

We use a multi-taper method (MTM) of spectral analysis (Huybers, 2021; Percival & Walden, 1993) to calculate
spectral power densities which are later converted to relative amplitudes within the 7-15 year band. MTM spectral
analysis averages multiple realizations of data using a series of Slepian tapers, offering a smoothed estimate of
prominent spectral characteristics. We spectrally analyze the EGRIP $\delta D$ record in 400-year windows with a timestep
(i.e. stepwise shift between adjacent windows) of 50 to 200 years, depending on desired temporal resolution. A
window size of 400 years provides adequate repetition of interannual to decadal signals for precise estimation of
high-frequency power density spectra.

## 2.3 High-Frequency Signal Attenuation

Firn is a porous and permeable substance comprised of snow and ice at the surface of an ice sheet. Water vapor within
interconnected firn pathways diffuses along temperature and concentration gradients. Isotopic exchange occurs at four
interfaces (i.e. (1) vapor-vapor, (2) vapor-ice surface, (3) ice surface-ice interior and (4) through the ice matrix), altering
the isotopic composition of deposited precipitation (Whillans & Grootes, 1985). Vapor phase diffusion

disproportionately dampens high-frequency signals but ceases below a pore density of 804 kg m$^{-3}$ (Johnsen et al., 2000). Solid phase diffusion continues below the pore close-off depth at a considerably slower rate (Jones et al., 2017). Spectral analysis is used to interpret the extent of vapor and solid phase diffusion to single frequency bands (Fig. A2). At EGRIP, the annual signal (i.e. 1 year) cannot be resolved, while the 2-5 year signals are significantly dampened throughout the LGP and Holocene. The 7-15 year band is well preserved throughout the entire EGRIP record, and is thus the focus of this analysis.

2.4 Gaussian Diffusion Correction

Values of cumulative mean water molecule diffusion, known as "diffusion lengths", can be estimated for windows of time or depth along a water isotope signal by evaluating dampened sections of its high-frequency spectrum (Gkinis et al., 2014; Hughes et al., 2020; Johnsen et al., 2000; Jones et al., 2017b; Jones et al. 2018; Jones et al. 2023; Kahle et al., 2021; Simonsen et al., 2011). Diffusion lengths are equivalent to the estimated standard deviation (i.e. sigma) of water molecule diffusion, which describes the statistical vertical displacement of water molecules from their original position (Fig. A3; Johnsen et al., 2000). The power density spectrum of a diffused water isotope signal, P(f), is defined as

$$P(f) = P_o(f) \exp\left[-(2\pi f \sigma_a)\right]^2 \qquad (1)$$

where $P_o(f)$ is the power density spectrum of the undiffused signal (in units of per mil$^2$ yr), f is frequency (in units of 1/yr), and $\sigma_a$ is diffusion length (in units of years). To solve for $\sigma_a$, we create twelve equally spaced logarithmic bins within the diffused interval and fit a Gaussian to P(f) using a least-squares fitting technique) (Fig. A4). The EGRIP $\delta$D diffused interval ranges between the 2-25 year band, and the Gaussian fit is optimized by manually adjusting this interval for each age window. This ensures the diffusion length is not calculated based on measurement noise, which occurs at an even higher frequencies and can be identified as a sudden bend in the slope of P(f). Diffusion lengths can be expressed in meters ($\sigma_z$) using the conversion:

$$\sigma_z = \sigma_a * \lambda_{avg} \qquad (2)$$

where $\lambda_{avg}$ is mean annual layer thickness (in units of m/yr). The diffusion corrected power density spectra, $P_o(f)$, is determined by rearranging equation (1), so that:

$$P_o(f) = P \exp\left[4\pi^2 f^2 \sigma_a^2\right] \qquad (3)$$

where P is the observed power density, f is frequency, and $\sigma_a$ is the solved diffusion length. We calculate average undiffused power density, $P_{ave}$, in the 7-15 year band by dividing the integral of power density by the frequency range (Jones et al. 2018):

$$P_{ave} = \frac{\int_{f_a}^{f_b} P_o(f)}{f_b - f_a} \qquad (4)$$

where $f_a$ and $f_b$ are the upper and lower limits of the frequency band, respectively. Relative strength ($P_{amp}$), or amplitude, is calculated for each window as the square root of average power density ($P_{ave}$):

$$P_{amp} = \sqrt{P_{ave}} \qquad (5)$$

190

### 2.5 Uncertainty

Diffusion length, $\sigma_a$, can be alternatively quantified as the slope, $m$, of a linear regression, $y$, fitted to ln[P(f)] versus $f^2$ of the diffused interval. Uncertainty bounds are defined as maximum and minimum slopes within one standard deviation of $y$ (Jones et al. 2017b, 2018) (Fig. A4).

195

### 2. 6 HadCM3

To investigate the physical processes underlying our results, we utilize the fully-coupled HadCM3 ocean-atmosphere general circulation model (GCM) to test how global climate responds to various forcing mechanisms in the North Atlantic (Gordon et al., 2000; Pope et al., 2000; Valdes et al., 2017). The simulations used here (Roberts and Hopcroft, 2020) force LGP stadial phases through the imposition of a freshwater flux or a perturbation to annual mean sea-ice cover. The hosing simulations impose freshwater uniformly across the North Atlantic between 50° to 70° N and include variations in flux between 0.04 and 1.0 Sv (10^6 m^3 s^-1). The perturbed sea-ice simulations artificially pin ice extent to a particular latitude between 40° to 70° N, though interannual ice edge variability is still possible due to advection in the module. Ice depth is prescribed to 4 meters and may reach a maximum concentration of 95%. To deconvolve how climate responds to both sea-ice and freshwater hosing, a forcing which combines the aforementioned simulations is included. To investigate a possible dependence on background climate state, we also impose the sea-ice and freshwater forcing onto boundary conditions simulating both the Last Glacial Maximum (LGM) and Pre-Industrial (PI). We calculate high-frequency, interannual-scale variability as the standard deviation of 7-15 year band pass filtered timeseries for temperature and sea-ice concentration output.

210

The freshwater hosing and pinned sea-ice simulations give a range of mean temperatures at the location of EGRIP that represent a spectrum of possible mean LGP climate states. By including a spectrum, we reduce the bias that could arise from defining a single stadial or interstadial state with only one forcing mechanism. Both forcings alter the distribution of sea ice, however the freshwater forcing also imposes a large, and potentially unrealistic, change to ocean circulation. Crucially, including these two sets of simulations controls for oceanic-driven effects when investigating the role of sea ice in driving climatic variability at EGRIP: if a response is general across all simulations, even for pinned sea-ice

simulations with no change in ocean circulation, we can infer that ocean circulation is not the immediate cause of the change.

220

## 3 Results

3.1 Interannual Water Isotope Variability

The raw EGRIP $\delta$D record exhibits millennial-scale D-O variability consistent with prior analyses of deep ice cores (e.g. GRIP, NGRIP) recovered from inland Greenland (Fig. 1a) (Greenland Ice-core Project (GRIP) Members, 1993; North Greenland Ice Core Project Members, 2004). Isotopic measurements range from approximately -240 to -310 per mil in the Holocene while the LGP is characterized by increased depletion and greater spread in values ranging from -260 to -380 per mil. On average, diffusion-corrected variability in the 7-15 year band is elevated during the LGP, 230 compared to the Holocene, reaching a maximum plateau between ~30-15 ka b2k. The average decrease in variability between the plateau and the Holocene is approximately 60% (Fig. 1b).

Large variations in 7-15 year variability during the LGP are evident. For example, the 7-15 year amplitude at ~38 ka b2k is so low that it is nearly the same magnitude as variability seen throughout the Holocene. Conversely, variability 235 at ~29.5 ka b2k is ~360% greater than in the Holocene. Another important detail is that the raw (i.e. non-diffusion corrected) 7-15 year variability record exhibits lower amplitudes, yet simultaneous shifts with the corrected record (Fig 1c). This demonstrates the ability of our correction to target signal attenuation by diffusion without incorporating uncertainty into the temporal evolution of the record. In other words, the signal we document is a robust feature of the climate system and not an artifact of the diffusion correction or laboratory analysis.

Within the context of the LGP, high-frequency variability is on average 18% lower in the 400 years following the onset of warm GI periods, relative to the onset of preceding cold GS periods (Rasmussen et al., 2014) (Fig. 2b). The largest reduction in variability from GS-onset to GI-onset is 37% for D-O Event 8, while the smallest change is indistinguishable from zero for D-O Event 9. In the case of D-O Event 9, the temporal length of the interstadial is only 245 260 years, which skews 400-year averages towards the background stadial state. There are no instances of greater variability during an GI-onset, relative to the prior GS-onset (Table 1). The above analyses demonstrate a negative correlation between water isotope ratios ($\delta$D) (which are a proxy for temperature) and strength (i.e. amplitude) of interannual and decadal isotopic variability in Northeastern Greenland throughout the last 50 kyr (Fig. 1c, 1d). It is important to note that we document multi-millennial excursions in variability occurring between 27-15 ka b2k wherein 250 cold GS conditions persist uninterrupted by abrupt warming, with the exception of D-O Event 2 around 23 ka b2k. The excursions are comparable, yet generally smaller in magnitude than those occurring between 50-27 ka b2k when D-O cycling is relatively consistent.

Next, we reduce the timestep of our 400-year window analysis from 200 to 50 years to analyze the lead-lag relationship between mean temperature change associated with D-O Events and isotopic variability (Fig. 3a, 3b). In most instances (i.e. D-O Events 2, 3, 4, 5.1, 5.2, 6, 7, 8, 9, 10, 12 13), we find that the reduction in 7-15 year variability from peak values initiates ~150-600 years **prior** to abrupt warming (Fig. 4). For these events, approximately 50% or more of the variability change has occurred before abrupt warming begins as defined by Rasmussen et al. (2014). There is only one case (i.e. D-O Event 11) in which the reduction in isotopic variability leads warming on sub-centennial scales. Due to our window size of 400 years, we can only conclusively say that lead times of > 200 years are significant beyond uncertainty of the diffusion correction. Thus, there is clear evidence that 7-15 year variability for D-O Events 2, 3, 4, 5.2, 6, 7, 8, 10, 12 and 13 leads centennial-scale mean temperature change at the onset of GI phases.

| | | | | | | D-O Event | | | | | | | |
|---|---|---|---|---|---|---|---|---|---|---|---|---|---|
| | **2** | **3\*** | **4\*** | **5.1\*** | **5.2** | **6\*** | **7** | **8** | **9\*** | **10** | **11** | **12** | **13** |
| **GS** | 133.2 | 132.4 | 157.2 | 126.5 | 141.8 | 106.5 | 124.6 | 106.5 | 94.5 | 117.5 | 118.1 | 113.4 | 101.6 |
| **GI** | 124.5 | 125.4 | 99.2 | 123.5 | 86.6 | 98.8 | 94.3 | 59.8 | 117.5 | 95.6 | 96.8 | 76.5 | 72.9 |
| **%** | 6.0 | 20.2 | 21.5 | 12.9 | 18.6 | 20.6 | 11.4 | 36.7 | -0.05 | 18.9 | 14.6 | 24.6 | - |

**Table 1 |** Percent (%) reduction in 7-15 year diffusion-corrected variability (per mil$^2$ yr) between Greenland Stadial (GS) onset and the following Greenland Interstadial (GI); D-O Events marked by an asterisk are characterized by interstadial phases shorter than the 400 year window size (240, 300, 240, 380, 260 years for D-O Events 3, 4, 5.1, 6, and 9, respectively); Due to the brief duration of G-I 2.1 and 2.2 (collectively referred to as D-O 2 in Table 1), one 400-year window is placed at the onset of G-I 2.2 (i.e. 23.34 ka b2k), which encompasses the entirety of G-I 2.2, a majority of G-I 2.1, and the short-lived stadial phase between each interstadial (Rasmussen et al., 2014)

3.2 HadCM3 Model Output

Interannual-scale (i.e. 7-15 year) temperature variability is assessed for Northeastern Greenland and the broader North Atlantic region. We find that temperature variability and mean temperature at the EGRIP site are negatively correlated (i.e. colder states are characterized by greater fluctuations in atmospheric temperature and vice versa) (Fig. 6b). These results agree well with our observations. To understand a wider regional context of temperature variability, we show a correlation map of 7-15 year variability across the North Atlantic relative to the 7-15 year variability at the EGRIP field site (Fig. 5a). Figure 5a shows the strongest remote correlation outside of Greenland exists in the North Atlantic subpolar gyre region, southwest of Iceland, and extending into the southernmost portion of the Norwegian Seas. Oceanic regions north of Iceland and east of Greenland (i.e. central Nordics Seas) show very low correlation (Fig. A6).

Next, we investigate the role of year-to-year sea-ice fluctuations on climate variability over Northeastern Greenland. Figure 5b displays the spatial correlation between 7-15 year sea-ice and temperature variability over Northeastern Greenland. Again, model output suggests ties between EGRIP and the North Atlantic, with the strongest correlations near 50° N 30° W. Figure 6a shows the individual simulation results at 50° N 30° W, with all simulations showing a

trend of increased EGRIP temperature variability with increased sea-ice variability. In the Nordic Seas, simulations exhibit little to no correlation between interannual-scale sea-ice fluctuations and temperature variability at EGRIP (Fig. A7). This finding is consistent with a persistent ice cover throughout both stadial and interstadial conditions (Li et al,. 2010; Hoff et al., 2016; Sadatzki et al. 2020), wherein ice variability is inherently low.

Correlation does not demonstrate causation, so we examine other factors which might explain our results. Changes in sea-ice variability might be associated with changes in mean temperature or with total sea-ice area. Therefore, we compare the strength of these variables in explaining trends of high-frequency temperature variability at the EGRIP location. Figure 6b shows how EGRIP temperature variability covaries with mean temperature. There is a strong negative correlation for all simulations, except those marked by grey crosses (Fig. 6b). This subset of simulations contains unrealistic sea-ice prescriptions (i.e. sea ice extends south of 50° N) (Brennan et al., 2013; Dokken et al., 2013; Sadatzki et al. 2020). The extended sea-ice cover causes a reduction in temperature variability that is greater than the reduced mean temperature would suggest; the grey crosses sit well below the regression line. Similarly, Figure 6c shows how EGRIP temperature variance covaries with total North Atlantic sea-ice area. We might expect that lower mean temperatures in Greenland are associated with extended sea-ice cover and it is therefore the mean sea-ice cover that affects EGRIP temperature variability. As with mean temperature, we find that the unrealistic sea-ice extent simulations fall far from the regression line. Furthermore, the correlation amongst realistic simulations is low ($R^2 = 0.45$), relative to other comparisons. Thus, mean temperature and sea-ice extent alone cannot explain shifts in temperature variability across LGP climate states. By contrast, a comparison between 7-15 year EGRIP temperature variability and North Atlantic sea-ice variability exhibits higher agreement ($R^2 = 0.60$) for all simulations (i.e. both realistic and unrealistic prescriptions); grey crosses fall on or near the regression line (Fig. 6a). This indicates a robust relationship and crucially, that 7-15 year sea-ice fluctuations in the North Atlantic are an excellent predictor of EGRIP temperature variability on similar scales.

**4 Discussion**

The results of this study are twofold. First, using HadCM3 we identify a critical coupling between interannual-scale North Atlantic sea-ice fluctuations and high-frequency temperature variability over Northeastern Greenland throughout the LGP. Conceptually, these results can be accounted for by our isotopic observations (i.e. stronger high-frequency variability during GS periods, and vice versa). Glacial stadial phases are characterized by lower average temperature and sea ice extending to the North Atlantic. It is possible that greater excursions in sea-ice concentration were driven by the enhanced latitudinal range between maximum summer and winter ice extents (Sadatzki et al. 2020), which presumably changed on a year-to-year basis. These seasonal variations would impart volatility in high-frequency temperature fluctuations on annual, interannual, and decadal scales via ice-atmosphere feedbacks. By extension, Rayleigh distillation and the isotopic signature of precipitation at EGRIP would also be impacted.

An additional contribution to enhanced isotopic variability during stadial phases may stem from altered source-to-sink pathways. With large seasonal swings in the capping and exposure of the ocean surface by sea ice, evaporative sources upstream of EGRIP would also be altered. As a consequence, variability in both evaporative source signatures and temperature gradients of moisture transport to Northeastern Greenland would increase. An isotope enabled GCM is required to test this hypothesis.

Our second, and key, finding is that reductions in 7-15 year isotopic variability generally begin hundreds of years prior to abrupt D-O warming at the EGRIP site. Taken together with results from HadCM3, we suggest that a fundamental change in North Atlantic sea-ice dynamics occurs in the centuries leading to Greenland warming. One possibility is that winter maximum ice extent retracted north of the high correlation North Atlantic pocket (50° N 30° W) in the centuries leading to abrupt D-O warming, thereby reducing the latitudinal range between seasonal extents and overall mean ice extent. Based on our results from HadCM3, such a change would also be correlated with lower 7-15 year temperature variability at EGRIP. Alongside this, greater exposed ocean surface in winter may increase winter temperatures, ultimately increasing mean annual temperatures at EGRIP. Some GS phases (i.e. prior to D-O Events 5.2, 8, and 10) exhibit a small rise in $\delta$D in the centuries prior to abrupt temperature increase. It is possible the GS phases which do not exhibit this behavior experience a muting of the $\delta$D increase due to source moisture deriving from more northerly, isotopically-depleted water as the ice edge recedes.

The following section outlines a framework in which this hypothesis fits within the existing literature. It has been previously suggested that a salt oscillator in the North Atlantic drove bimodal shifts in AMOC and hence, transitions between stadial and interstadial conditions during the LGP (Broecker et al., 1990; Peltier & Vettoretti, 2014). Using HadCM3, Armstrong et al. (2022) demonstrates a complex ocean-atmosphere-ice feedback capable of maintaining a millennial-scale salt oscillation mechanism. To summarize, the salinity gradient between the tropics and North Atlantic regulates advection between these regions, thus governing the strength of AMOC, northward heat transport and related regional climate effects. In the model, the onset of cold stadial periods are marked by a sluggish AMOC, accumulating salinity in the tropics and diminishing salinity in the North Atlantic. Due to a reduction in the transport of dense (i.e. salty) and warm water north, deep-water formation drops to a minimum. Concurrently, the difference in PSU (i.e. Practical Salinity Unit) between the tropics and North Atlantic approaches a maximum. It is precisely this feature, peak salinity gradient, that kickstarts AMOC and initiates a transition from stadial to interstadial in the salt oscillator framework.

Sea ice and surface freshwater fluxes are deemed critical in developing the salinity gradient, though their effects are spatially heterogeneous. In the North Atlantic, both stadial and interstadial periods are characterized by a freshening effect. Winter ice traps freshwater by freezing sea water and trapping overhead precipitation. The summer melt season flushes out both freshwater sources, resulting in a net freshening effect. Due to reduced ice extent during interstadials, this freshening effect is dampened. A relevant detail of the salt oscillation mechanism is that North Atlantic salinity

and AMOC strength begin to steadily increase partway through the stadial phase (i.e. centuries prior to interstadial onset), priming the system for an abrupt shift. Critically, Armstrong et al. (2022) proposes reduced regional freshening due to gradual sea-ice retreat as a likely cause. Our study provides paleoclimate and modeling evidence in favor of this suggestion.

A useful tool in corroborating the centennial-scale phase offset is empirical evidence of North Atlantic sea ice behavior with similar lead-lag behavior. Unfortunately, high-resolution records are sparse. One such analysis (Sadatzki et al., 2020) finds declining seasonal sea ice in the lower latitude Norwegian Sea (MD99-2284; 62° N, 0° W) initiated ice-free conditions further north (MD95-2010; 66° N, 04° E) prior to Greenland temperature increase. In this study, sediment core age models are tied to GICC05 using stratigraphic alignment of ARM (anhysteretic remanent magnetization), near-surface temperature, and cryptotephra layers with NGRIP $\delta^{18}O$. At the northern core site, a change in the properties of biomarkers Brassicasterol and HBI-III, occurring in open-ocean environments, is detectable centuries prior to the abrupt GS-GI transition. A direct comparison to our isotope variability timeseries shows near simultaneous shifts, possibly indicating a common link (Fig. 7). However, the MD95-2010 sediment core site falls just outside the high-correlation pocket suggested by HadCM3. One explanation is that large scale changes in Norwegian Sea mean ice extent are not associated with shifts in variability, unlike the North Atlantic. It is also possible the model grid cells are too coarse to resolve fine detail in Norwegian Sea ice movement, thus failing to detect shifts in variability as mean sea-ice area changes. Nevertheless, this realization prompts interest in high-resolution sea-ice proxy records at the heart of the high variance correlation.

Lastly, an inexplicable component of this study is the continuation of large excursions in high-frequency isotopic variability even when D-O cycling is turned off for long stretches (i.e. 27-15 ka b2k). In some cases, the fluctuations are comparable in magnitude to those occurring across prior GS-GI transitions. It seems a climate variability oscillation is inherent to the LGP background state, yet does not result in abrupt mean climate change (i.e. D-O Events) based on certain boundary conditions which remain to be seen. Due to the simultaneous occurrence of the Last Glacial Maximum during this timeframe, an obvious factor to test in future studies is the height and extent of the Laurentide and Scandinavian Ice Sheets and their effects on climate variability.

There are additional explanations of the lead-lag result, though they currently lack strong evidence. The first variable to consider is stratigraphic noise, which is non-climatic variability imparted to the water isotope record due to processes like precipitation intermittency, surface sublimation, and wind-driven snow erosion (Fisher et al., 1985; Helsen et al., 2005; Town et al., 2008; Zuhr et al., 2021). Stratigraphic noise hinders the extraction of climate-induced high-frequency signals during low accumulation phases (e.g. LGP stadials), thus raising concerns that local depositional processes may also drive the results presented in this study. Unfortunately, the temporal evolution of stratigraphic noise cannot be quantified directly and currently, there are no LGP signal-to-noise ratio comparisons with nearby Greenland ice cores. Still, contributions of non-climatic noise are likely state dependent (e.g. GI vs GS phases) and inherently linked to

accumulation rate. EGRIP 7-15 year variability also exhibits a centennial lead-lag with accumulation, suggesting the primary driver for this deviation lies elsewhere (Fig. A8).

Second, water-isotope diffusion is a property not exclusively set at the ice sheet surface, but one that reflects the duration of the firn densification process. If the surface climate changes, this impacts concurrent precipitation in addition to the firn column below via temperature gradients, which are important for grain metamorphosis and vapor movements via overburden pressure and barometric pumping. Changes in interannual-to-decadal variability in ice that is deeper (i.e. older) than abrupt D-O warmings may be driven by shifting thermal gradients that affect grain metamorphosis and vapor transport in such a way to keep pore pathways open longer, thereby enhancing gas mixing and ultimately reducing water isotope variability in older ice. Such a mechanism is not currently captured in accepted firn models (e.g., Johnsen et al, 2000).

As an analysis of shifting thermal gradients based on results from the WAIS Divide ice core in West Antarctica, both diffusion lengths (Jones et al. 2017b) and total air content (TAC) (Buizert et al., 2021) exhibit anomalous increases around 19 ka b2k, as temperatures rose during deglaciation. TAC may reflect thermal gradients, overburden pressure, or barometric pumping, but may also be sensitive to surface climate drivers. Efforts to model the anomalous WAIS Divide period have been mostly unsuccessful (Jones et al. 2017b). The increase in diffusion at WAIS Divide was not accompanied by any reductions in interannual-to-decadal isotopic variability (Jones et al. 2018), which dramatically lowered 3 thousand years later when diffusion lengths and TAC decreased, the exact opposite of what might be expected based on the above explanation. There are further complicating factors with a thermal gradient-firn metamorphosis hypothesis. Figure A9 shows that multi-decadal variability (20-30 years) in EGRIP water-isotopes – frequencies that should have nearly zero impact from firn diffusion - also exhibit a centennial-scale lead time in line with the higher frequency and more diffused 7-15 year band. This suggests that 1) firn processes that affect diffusion, like thermal gradients, cannot be a driving factor because the entire spectrum of variability (both diffused and non-diffused frequencies) is altered or 2) our current physical understanding of diffusion is substantially wrong and highly non-Fickian. Further, there are large changes in interannual to multi-decadal variability between 27-15 ka b2k (up to 40% reductions) when DO cycles are infrequent, and hence no dramatic thermal shifts are imparted to the firn column.

As a final note, evidence of centennial-scale lead times in sea-ice proxies from Norwegian sediment cores as compared to Greenland abrupt warming, as well as the model evidence presented here, suggest the weight of evidence is currently in favor of a regional climate driver rather than firn dynamics or stratigraphic noise. However, if new evidence arises in the future (e.g. improved firn modeling, signal-to-noise ratio analysis across multiple deep Greenland ice cores) our working hypothesis will need to be modified.

**5 Conclusion**

Interannual and decadal variability is an important aspect of paleo-climate reconstructions and assessments of D-O warming, but such records have been previously sampled at lower resolution, which may exclude important climate

insights that are documented here. In this study, the mm-scale EGRIP water isotope record is used to quantify diffusion-corrected 7-15 year isotopic variability for Northeastern Greenland to 49.9 ka b2k. Importantly, we provide evidence that high-frequency LGP Greenland climate variability became decoupled from local mean temperature in the centuries preceding D-O warming. Our initial model analysis suggests lower latitude feedbacks associated with interannual North Atlantic sea-ice change may be a driver of this offset. As the cause of D-O cycles continues to be debated, these results should be considered.

Additionally, future isotope-enabled GCM studies may benefit from utilizing the high-frequency EGRIP variability timeseries, presented here, to constrain boundary conditions or benchmark model output. We suggest targeted tests aimed at temporally reconciling the centennial-scale offset with sea-ice behavior to better understand regional North Atlantic climate change within the context of abrupt D-O warming. Analysis of high-resolution sediment proxy records from critical locations identified in this study may also clarify uncertainties.

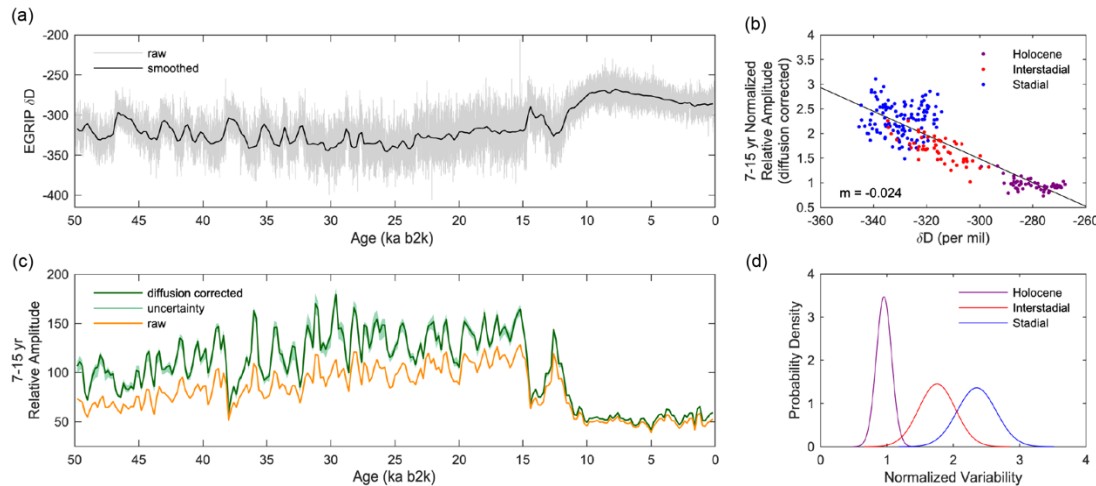

**Figure 1 | (a)** EGRIP δD to 50 ka b2k (grey) overlaid by down-sampled data for visual clarity (black; window = 400 years; timestep = 200 years); **(b)** diffusion-corrected data from panel (c) binned by climate state and fit with linear regression ($r^2 = 0.75$; $p < 0.05$); **(c)** Raw (orange) and diffusion-corrected (green) strength of 7-15 years isotopic variability at EGRIP (window = 400 years; timestep = 200 years); **(d)** variance in binned data from panel (b)

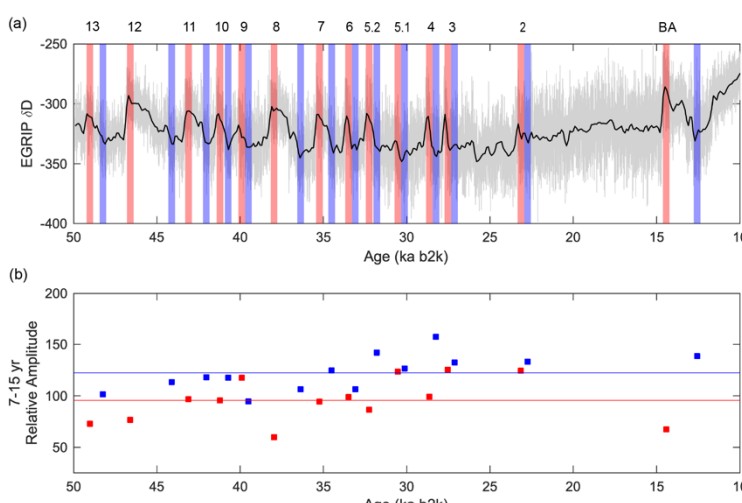

**Figure 2 | (a)** EGRIP δD overlaid by 400-years windows at onset of Greenland Interstadial (GI; red) and Greenland Stadial (GS; blue) periods (Rasmussen et al., 2014); For GI phases of less than 400 years, the onset of the subsequent GS phase is defined as the termination of the GI window to avoid overlap; Due to the brief duration of G-I 2.1 and 2.2, one 400-year window is placed at the onset of G-I 2.2 (i.e. 23.34 ka b2k), which encompasses the entirety of G-I 2.2, a majority of G-I 2.1, and the short-lived stadial phase between each interstadial; D-O Events are indicated on upper x-axis **(b)** diffusion-corrected 7-15 year variability of windows outlined in panel (a)

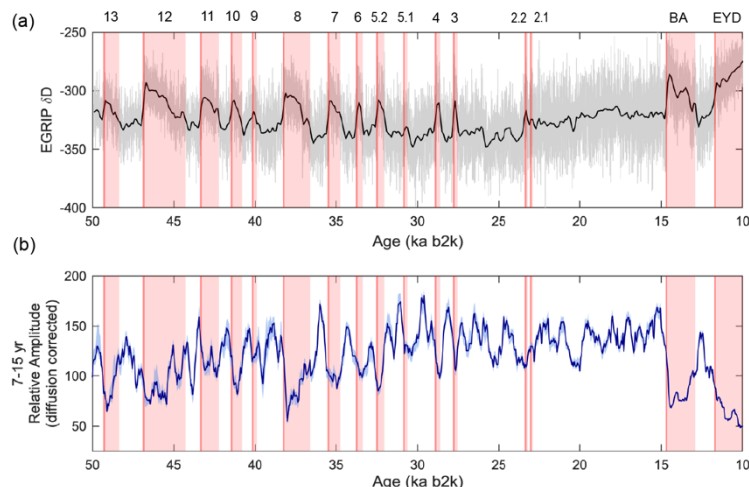

**Figure 3 | (a)** EGRIP δD (black; window = 400 years; timestep = 50 years) shaded by duration of warm Greenland Interstadial periods 2-13, the Bølling-Allerød (BA), and End Younger Dryas (EDY) (red; Rasmussen et al., 2014); **(b)** high-resolution analysis of 7-15 year isotopic variability (dark blue; window = 400 years; timestep = 50 years) with uncertainty bounds (light blue); D-O Events are indicated on upper x-axis

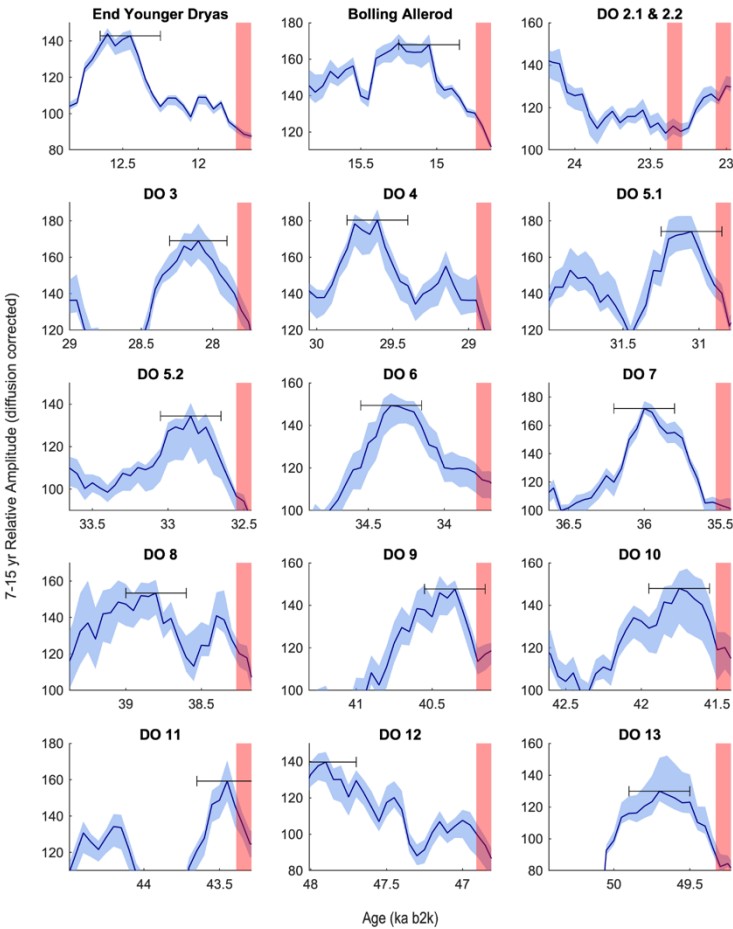

**Figure 4 |** Zoomed view of GS-GI transitions from Figure 3 panel (b) for D-O Events 2-13, the Bølling-Allerød (BA), and End Younger Dryas (EDY); high-resolution analysis of 7-15 year isotopic variability (dark blue; window = 400 years; timestep = 50 years) with uncertainty bounds (light blue); black horizontal error bar indicates 400-year window range at height of variability; red shading represents 50-year buffer before and after onset of GS-GI transition as defined in Rasmussen (2014)

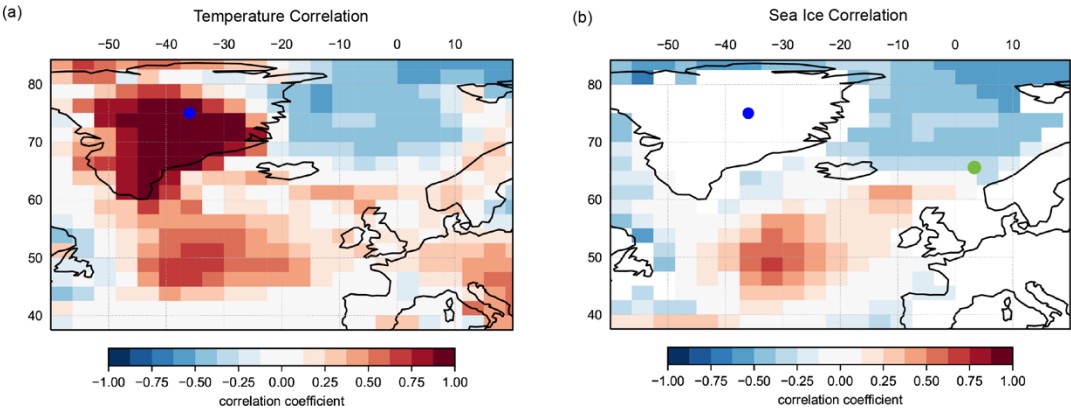

**Figure 5 | (a)** Correlation between 7-15 year temperature variability at EGRIP (blue dot) and other regional grid cells; **(b)** Correlation between 7-15 year temperature variability at EGRIP (blue dot) and 7-15 year sea ice variability for all regional oceanic grid cells; red and blue shading depicts high and low correlations, respectively; MD95-2010 core site is marked by a green circle

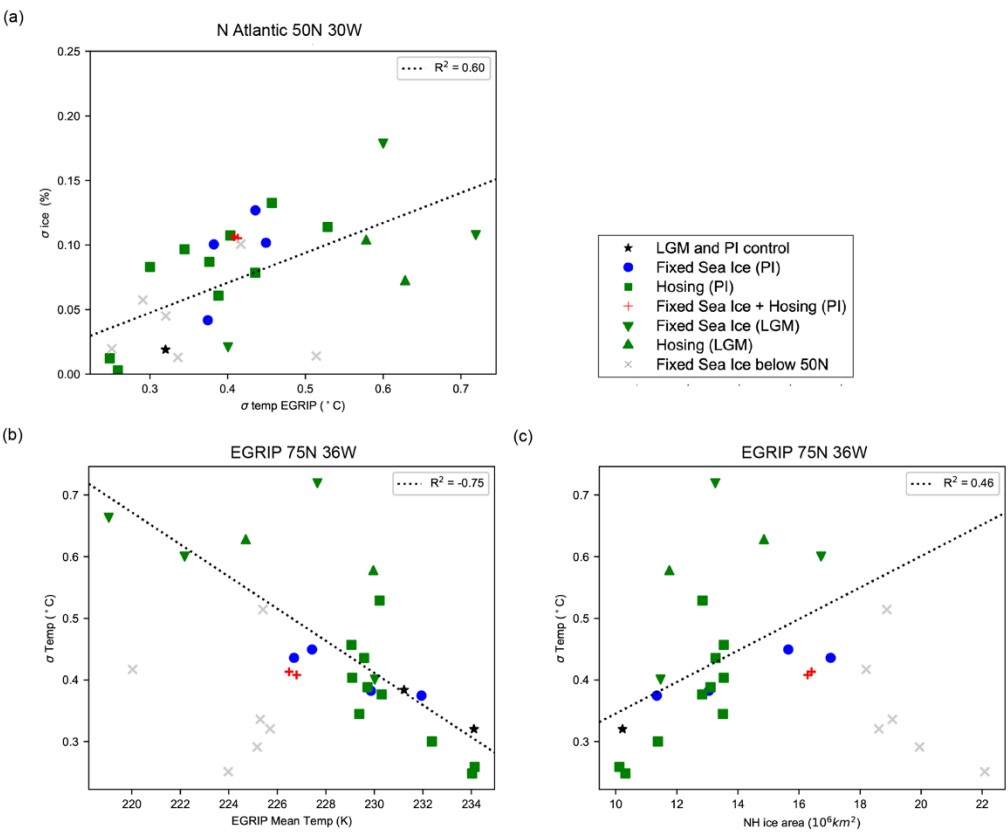

**Figure 6 |** Individual model results for **(a)** 7-15 year sea ice variability versus 7-15 year EGRIP temperature variability near the high-correlation North Atlantic pocket (i.e. 50°N 30°W) **(b)** 7-15 year temperature variability versus mean temperature at the EGRIP location **(c)** 7-15 year temperature variability versus Northern Hemisphere ice area at the EGRIP location; Note: simulations including sea ice extending below 50° N (grey x) have been excluded from regression

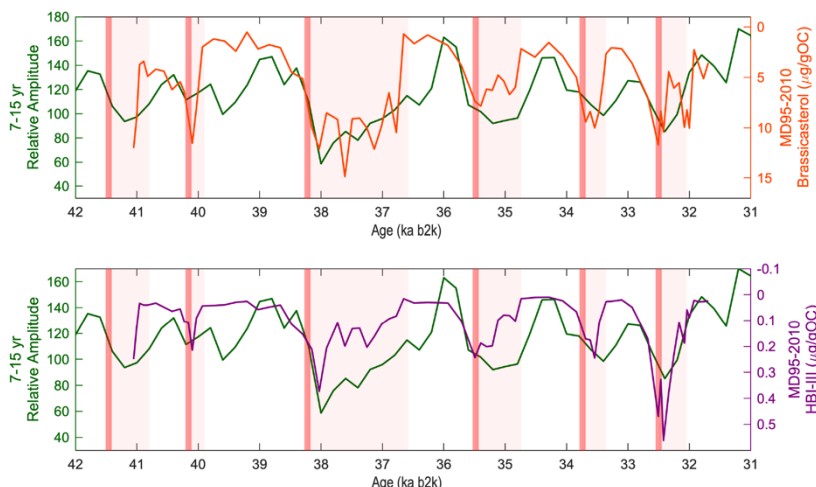

**Figure 7** | Comparison between 7-15 year diffusion correct isotopic variability at EGRIP and open ocean biomarkers (i.e. HBI-III and brassicasterol) recovered from ocean sediment at 66° N, 04° E; dark red shading represents 50-year buffer before and after onset of GS-GI transition; light red shading represents duration of GI period as defined in Rasmussen (2014)

**Appendix A: Figures**

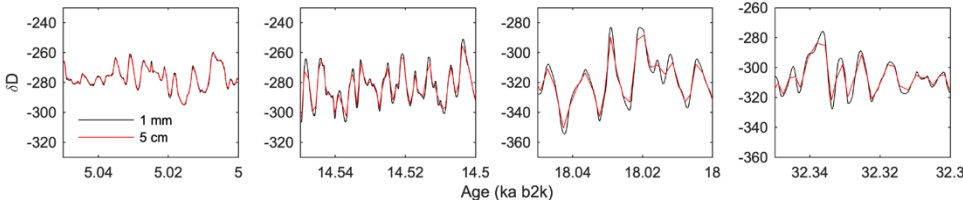

**Figure A1 |** Comparison of original 1 mm-EGRIP $\delta$D (black) to downsampled 5-cm EGRIP $\delta$D (red); Each window contains 50 years of data with progressively older windows towards the right; lower sampling resolution artificially reduces the amplitude of interannual and decadal fluctuations after approximately 12 ka b2k

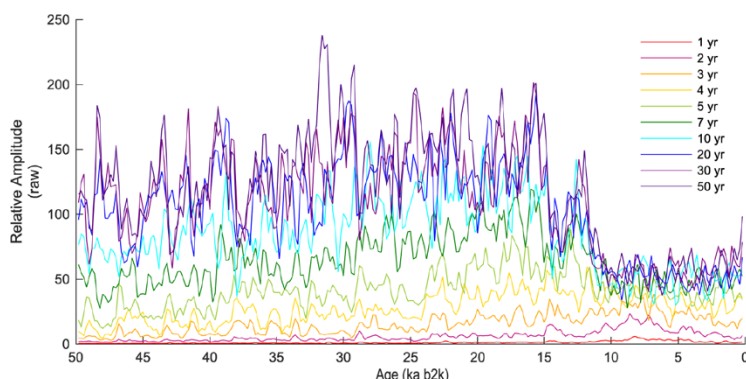

**Figure A2 |** Strength of raw (i.e. non-diffusion corrected) isolated frequencies in the EGRIP $\delta$D record; Substantial attenuation with age due to diffusion in the firn layer is evident in the 1-5 year signals

**Figure A3 |** Comparison of diffusion lengths calculated using the EGRIP $\delta$D record in the time domain (black) versus the depth domain converted to the time domain using annual layer thickness (red); uncertainties largely overlap with some deviations prior to 43 ka b2k; Inverted EGRIP $\delta$D (blue) is plotted to show the synchronous shifts of diffusion length with mean climate state.

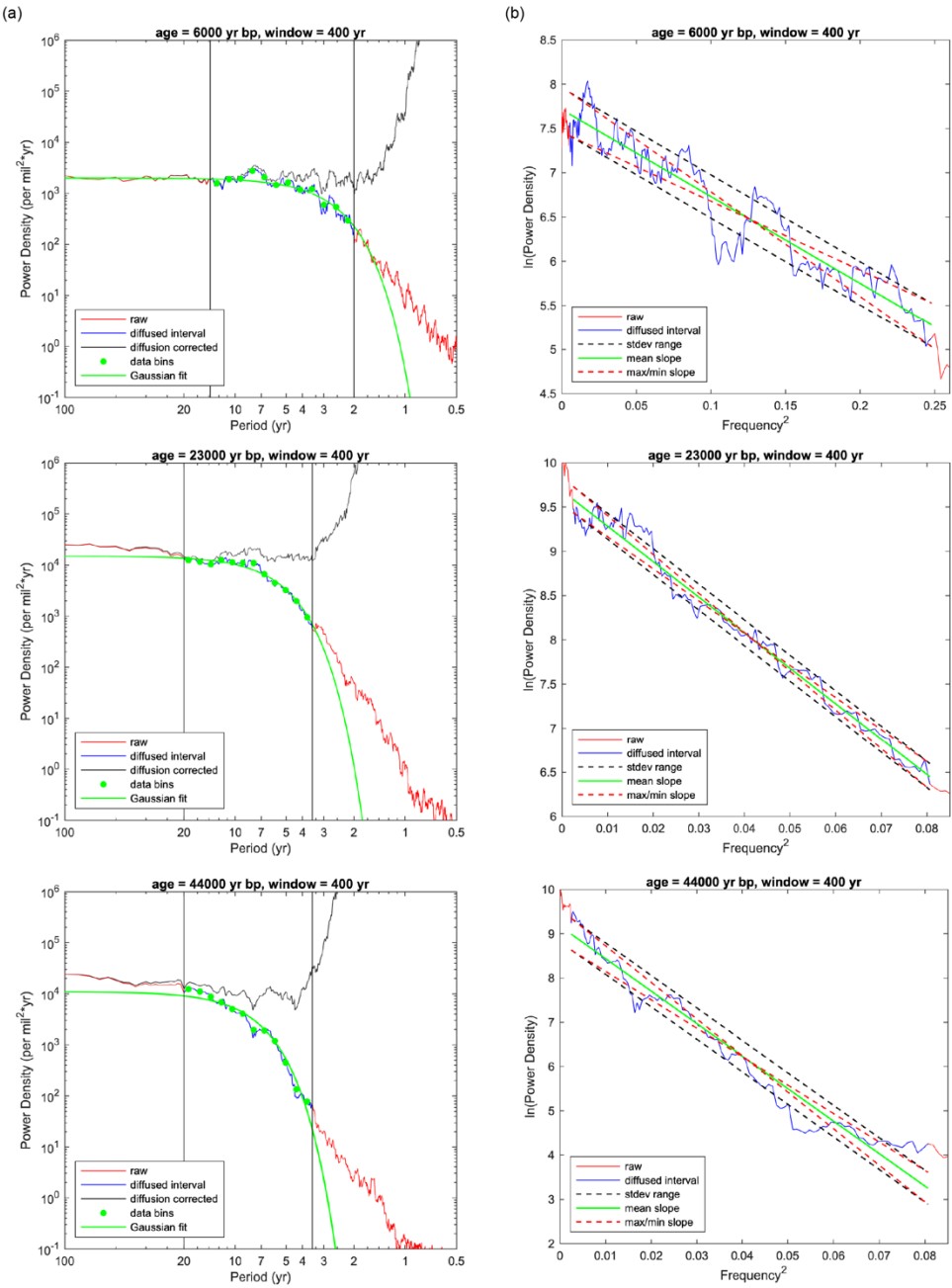

**Figure A4 | (a)** Examples of 400-year power density spectra (PSD) fitted by Gaussian regression for diffusion correction (progressively older windows towards bottom of panel); Twelve equally spaced logarithmic bins are used in fitting so that higher frequencies with increased data density do not skew correction **(b)** Examples of uncertainty calculation; maximum and minimum slopes of [ln(PSD) versus frequency$^2$] within one standard deviation represent
the high and low uncertainty bounds, respectively

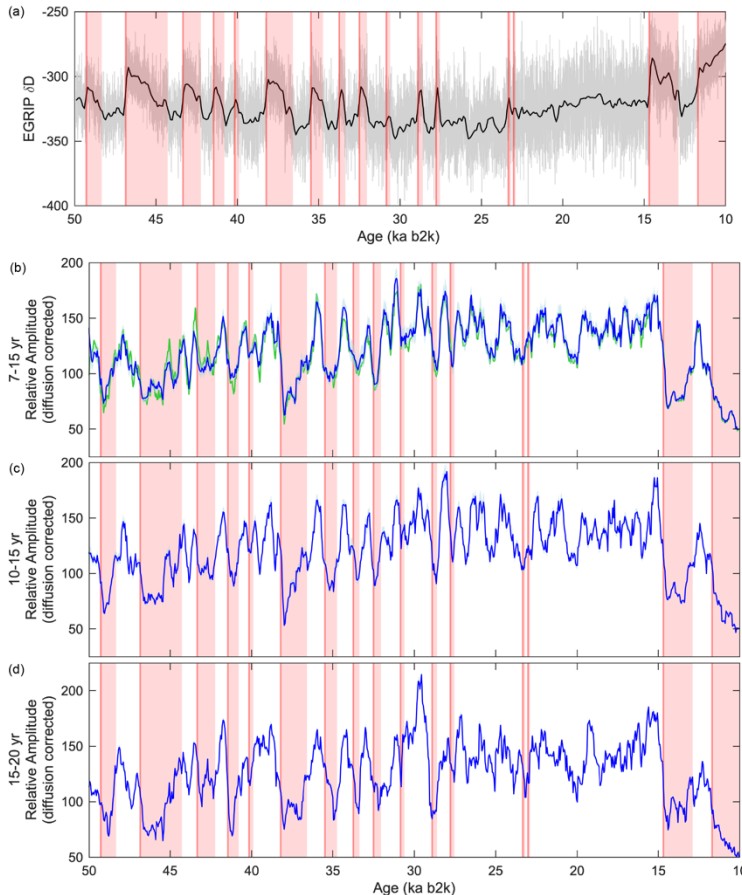

**Figure A5 | (a)** EGRIP δD (black; window = 400 years; timestep = 50 years) shaded by duration of warm Greenland Interstadial periods 2-13, the Bølling-Allerød, and Holocene (red; Rasmussen et al., 2014); High-resolution analysis of **(b)** 7-15, **(c)** 10-15 and **(d)** 15-20 year EGRIP isotopic variability (blue; window = 400 years; timestep = 50 years) using diffusion lengths which have been calculated in the depth domain and converted to the time domain using annual layer thickness; Results from Fig. 3 (green) are plotted underneath results in panel **(b)** to demonstrate that the method of diffusion-length calculation does not affect the lead-lag relationship between variability and the mean

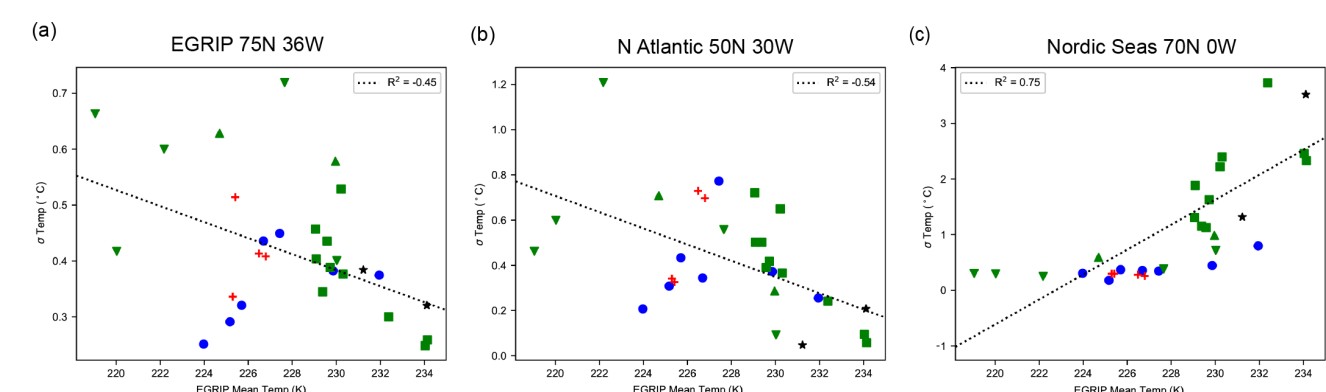

**Figure A6 |** Individual simulation results for the correlation between 7-15 year temperature variability and EGRIP mean temperature at (a) EGRIP, (b) North Atlantic, and (c) Central Nordic Seas

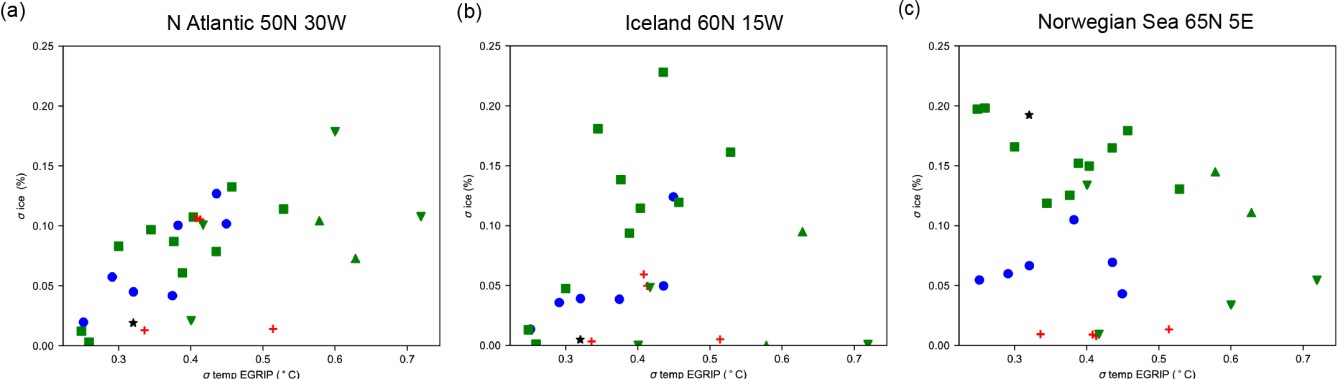

**Figure A7 |** Individual simulation results for the correlation between 7-15 year sea ice variability and 7-15 year EGRIP temperature variability at (a) North Atlantic, (b) Iceland and (c) Norwegian Sea

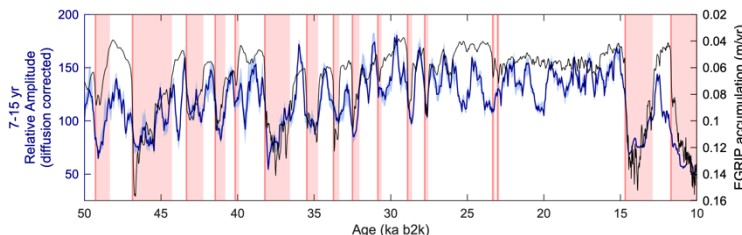

**Figure A8 |** Coevolution of high-resolution 7-15 year isotopic variability (dark blue; window = 400 years; timestep = 50 years) with inverted EGRIP accumulation (black)

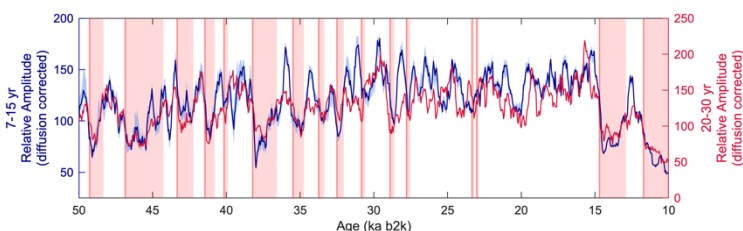

**Figure A9 |** Coevolution of EGRIP high-resolution 7-15 (dark blue; window = 400 years; timestep = 50 years) and 20-30 year isotopic variability (red; window = 400 years; timestep = 50 years)

**Date Availability.** EGRIP $\delta D$ and $\delta 18O$ data to 49.9 ka b2k is available through the NSF-Arctic Data Center (doi:10.18739/A20K26D2M)

**Author Contributions.** CAB and TRJ contributed to all aspects of this paper. TRJ designed the study. BHV, VM, TRJ and JWCW developed the continuous flow analysis system. WHGR provided HadCM3 model output. SOR developed the age-depth scale. BMV and BHV co-led the EGRIP water isotope consortium. VM, BHV, RN, TRJ, CAB, KSR, AGH, WBS, VG, CH, MFK, SEK, PL, FM, JR, MS, GS, and SSJ measured the high-resolution EGRIP water isotope data. KMC and TRJ developed the spectral techniques for diffusion correction. CAB wrote this article with significant editorial contributions from TRJ and comments from all authors. TS, CB, BHV, and JWCW were principal investigators on this project, funded through the National Science Foundation.

**Competing Interests.** At least one of the (co-)authors is a member of the editorial board of Climate of the Past.

**Acknowledgements.** The East Greenland Ice Core Project (EGRIP) was facilitated by collaborations between the Danish Centre for Ice and Climate, the US Office of Polar Programs (OPP), the National Science Foundation (NSF) and U.S. Air National Guard. The authors acknowledge scientific and logistical support from the entire EGRIP community.

**Financial Support.** This research has been supported by the National Science Foundation (grant no. 1804154 and 1804133) and the Niels Bohr Institute at the University of Copenhagen. SOR and GS acknowledges support from the Carlsberg Foundation (ChronoClimate) and the Villum Investigator Project IceFlow (grant no. 16572).

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
