# Peer review of "Shifts in Greenland interannual climate variability lead Dansgaard-Oeschger abrupt warming by hundreds of years"

_EGUsphere, 2024_

## Author Comment (AC1)

**Summary**

An ultra-high resolution time series for central/eastern Greenland deuterium variability is presented. After correction for diffusion, 7-15-year variability is considered as represented and and interpreted in terms of temperature variability that changes between Greenland stadials and interstadials. Temperature change at the ice core site on decadal scales is compared to sea-ice and sea surface-temperature change in the North Atlantic, and interpreted as primarily arising from sea-ice variability.

Crucially, the authors suggest that a reduction of decadal temperature variability at the ice core site occurs centuries prior to the interstadial warming, and corroborate this with the analysis of model simulations and one high-resolution marine core.

The study is well-written, and the topic is relevant to Climate of the Past. However, there are a few points were more detail, or more stringent acknowledgement of uncertainties, is important.

**Major Points**

- Please acknowledge the assumption of the validity of the temperature interpretation of dD at sub-decadal timescales -- as you show, sea-ice variability is highly correlated with temperature, but it is not the only driver.

    - The authors acknowledge uncertainties in translating the water isotope record directly to temperature, especially at decadal and interannual scales. It is also acknowledged that further isotope modeling studies are required, as the current understanding of water isotopes at high-frequency scales is limited. The following lines highlight this:

        - Line 48: "Stable isotopes of hydrogen ($\delta$D) and oxygen ($\delta^{18}$O) in polar ice cores provide information about local temperature and atmospheric circulation (Dansgaard, 1964)"

        - Line 303: "It is possible that greater excursions in sea-ice concentration were driven by the enhanced latitudinal range between maximum summer and winter ice extents (Sadatzki et al. 2020), which presumably changed on a year-to-year basis. These seasonal variations would impart volatility in high-frequency temperature fluctuations on annual, interannual, and decadal scales via ice-atmosphere feedbacks. By extension, Rayleigh distillation and the isotopic signature of precipitation at EGRIP would also be impacted. An additional contribution to enhanced isotopic variability during stadial phases may stem from altered source-to-sink pathways. With large seasonal swings in the capping and exposure of the ocean surface by sea ice, evaporative sources upstream of EGRIP would also be altered. As a consequence, variability in both evaporative source signatures and temperature gradients of moisture transport to Northeastern Greenland would increase. An isotope enabled GCM is required to test this hypothesis."

- One additional concern is that the isotopic signal we document may also be related to non-climate noise at high-frequencies, which changes across climate states (glacial interstadials vs stadials). It has been proposed that shifting accumulation rates may cause such a systematic influence on non-climate noise, as colder periods are associated with lower accumulation and thus greater precipitation intermittency and stratigraphic noise. To refute this, the authors have also now included the coevolution of EGRIP accumulation with 7-15 diffusion corrected variability in Figure A8. It shows that declines in variability lead accumulation shifts by hundreds of years for most D-O events. This suggests local depositional effects during cold stadial phases cannot account for the early shifts in 7-15 year variability, relative to D-O warming.

- Consider, in the Discussion, the robustness of the model-based process interpretation considering a usefulness of model intercomparison (e.g., [1]), in particular given the complexity of sea-ice models. Similarly, the marine record is a (hand-picked) example, there would be potential for targeted synthesis work here.

  - The author is waiting for input from coauthor before addressing this comment.

- The figures need to be reworked. Standard red/green looks grey to quite a few people (Listen, e.g. to these people describing their experience https://www.youtube.com/watch?v=FKSOe5NK_qQ and imagine how distinguishable colors are in most of your figures...)

  - Figures 1, 3 and 4 have been reworked to remove all standard red/green color combinations

**Minor Comments**

- p1l26/27 This sentence is ambiguously phrased. (Why) should there be a phase offset, and should it be distinguishable? I read this as: "Across stadial/interstadial transitions proxy evidence showed in-phase changes in mean temperature/dust/sea-salt concentration/accumulation rate".

  - The author is stating (not questioning) that proxy evidence (mean temp/dust/sea salt/accumulation) exhibits in-phase shifts across stadial-interstadial transitions consistent with the cited literature (Capron et al., 2021).

- p1l29/30 You write that high-frequency interannual variability surrounding "mean temperature change" has not been investigated -- please clarify that by mean change you mean centennial-to-millennial scales, and by high-frequency interannual variability. From a lower-resolution marine point of view both timescales are "just" variability.

  - This has been rephrased

- p2l71 "tipping-point sea-ice displacement" - The concept of a threshold, below which the ice edge becomes unstable, and fast/complete retreat of perennial sea-ice cover occurs is debated [2,3]. Compared to the ice sheet, sea-ice itself has little memory, but small changes in the ice edge may lead to large impact warming. Please rephrase.

  - The author has removed the phrase "tipping-point"

- p3l85 and following: Investigating leads and lags, as well as interannual to multicentennial variability across the LGP and for different timescales ("mean" vs. "variability") and attributing it primarily to local temperature change assumes that dD is faithfully representing local EGRIP site temperature. This is not explicitly mentioned, but is permeating the study, and should be acknowledged explicitly. Sea-ice variability, independently of temperature change at the EGRIP site, can induce d18O variability [4], and, as the authors themselves show with the model-based correlations these variables are colinear. Isotope-enabled simulations could allow, to some extent, to disentangle these relationships.

  - See comment under first bullet point of "Major Points". The authors do not intend to assume dD faithfully represents local EGRIP site temperature and present multiple arguments backed by their results on why the water isotope record at interannual and decadal scales is largely uncertain at present. The authors suggest isotope enabled modeling in future work.

- p4l141 timestep of 50-200 years -- presumably these are the shifts for the moving windows? Unclear.

  - Yes, this has been clarified in the manuscript on Line 144: "We spectrally analyze the EGRIP $\delta$D record in 400-year windows with a timestep (i.e. stepwise shift between adjacent windows) of 50 to 200 years, depending on desired temporal resolution"

- p5l154 correct: preserved

  - This has been corrected

- p6l204: instead of "spectrum of change" suggest rephrasing

  - This has been rephrased. Line 206 now reads: "The freshwater hosing and pinned sea-ice simulations give a range of mean temperatures at the location of EGRIP that represent a spectrum of possible mean LGP climate states. By including a spectrum, we reduce the bias that could arise from defining a single stadial or interstadial state with only one forcing mechanism."

- p7l217 increased depletion

  - This has been corrected

- p8l251 clarify "mean" timescale (see above)

  - This has been addressed. Line 259 now reads: "Thus, there is clear evidence that 7-15 year variability for D-O Events 2.2, 3, 4, 5.2, 6, 7, 8, 10, 12 and 13 leads centennial-scale mean temperature change at the onset of GI phases.

- p8l255 and following: How long are the simulations? How are the degrees of freedom and a significance for the correlations calculated? Are these step-wise simulations, and is the mean change then subtracted prior to correlation? The magnitude of the correlation is surprisingly high.  To what extent are these correlations representative for other models (given the fairly simple sea-ice model in HadCM3)?

- All simulations are run for 500 years, branched from a control simulation (Pre-industrial/LGM) that has been run for many 1000s of model years. Climatologies are computed from the last 100 years. All simulations are self contained, independent, simulations and branch from the same control simulation at the same time - they are not "step-wise" in the sense that the simulation with the sea ice edge at 55N feeds the simulation with the sea ice edge at 50N etc. In fig 5 (a) and (b) no means are subtracted as the correlation is between the variability (variance) in sea ice concentration and temperature. In figs 6(b) and (c) the abscissae do not have the control mean temperature removed: we wish to establish the relationship between cold and warm states and variability, therefore must include the overall colder conditions that arise from the LGM climate. The correlations in fig 5(a) are indeed high as in this figure we are correlating temperature variability with itself - over EGRIP this is by definition 1. This map gives a sense of the spatial autocorrelation for 7-15 year variability over the North Atlantic. Unsurprisingly over Greenland this is high. The correlations with sea ice are maximum at around 0.6, so sea ice variability can explain ~30% of the variance at EGRIP, thus the majority of the variance arises from other sources. Since no other model has been run across such a range of different forcings, it is impossible to say to what extent the results arise from HadCM3's quirks. However, over the 7-15 year timescale the atmosphere has little memory so coherent changes are driven by the ocean/ice system. Therefore, the magnitude of the correlations that we see arise from the dynamics of the atmosphere. Thus the relatively simple sea ice model in HadCM3 is not likely to bias the results.

- p10l350 Arguably, this is a single core site for which a reduction of sea-ice occurs prior to Greenland isotopic/temperature change, and a single climate model. The correlation patterns of sea-ice variability with EGRIP temperature in other models would be interesting. What is the age model of MD95-2010 based on, and what is the corresponding age uncertainty? Hopefully (or perhaps, evidently, from the results) not tie-points to GICC05. Perhaps this is an age model issue?

  - The authors acknowledge further isotope-enabled modeling and high-resolution sea ice reconstructions are critical (Line 383): "Additionally, future isotope-enabled GCM studies may benefit from utilizing the high-frequency EGRIP variability timeseries, presented here, to constrain boundary conditions or benchmark model output. We suggest targeted tests aimed at temporally reconciling the centennial-scale offset with sea-ice behavior to better understand regional North Atlantic climate change within the context of abrupt D-O warming. Analysis of high-resolution sediment proxy records from critical locations identified in this study may also clarify uncertainties."

  - Text regarding sediment core age model taken directly from Sadatzki et al., 2020:

    - "Accordingly, the age model of MD99-2284 is based on alignment of near-surface temperature signals and D-O climate transitions, independently verified and constrained by four distinct cryptotephra layers that were identified before and after the GS6GI5 transition as well as during GI6 and GI8 in both core MD992284 and the NGRIP (North Greenland Ice Core Project) ice core with a consistent geochemistry (33)

(SI Appendix, Materials and Methods and Fig. S2). Moreover, the glacial sediment sections in both cores MD95-2010 and MD99-2284 reveal a very consistent variability in anhysteretic remanent magnetization (ARM), reflecting deep ocean circulation changes in the Nordic Seas (13, 34, 35), which closely resemble the D-O climate fluctuations recorded by the δ18O of the NGRIP ice core. This enables development of an age model for core MD95-2010 by stratigraphic alignment of its ARM record to that of MD99-2284 and the δ18O of the NGRIP ice core (SI Appendix, Materials and Methods and Fig. S2). Thereby, our sedimentary sea ice records are placed on the Greenland ice core chronology GICC05 (36) and can thus be directly compared with the RECAP sea ice record, which also has been transferred to the GICC05 chronology by alignment of the RECAP dust record to NGRIP δ18O".

- o The above text has now been summarized in the manuscript on Line 355: "In this study, sediment core age models are tied to GICC05 using stratigraphic alignment of ARM (anhysteretic remanent magnetization), near-surface temperature, and cryotephra layers with NGRIP $\delta^{18}O$."

- Fig. 3, 4, 6, 7 please avoid red/orange and green as dominant colors in figures (not colorblind friendly)

  - o This has been addressed

- Fig. 5, perhaps add the mean position of the sea-ice edge in these figures for the LGM to aid interpretation.

  - o The author does not think this information would significantly aid interpretation as ice extent can be inferred from mean temperature in Figures A6 and A7, which are used to create the plots in Figure 5. Further, we use ~20 simulations over a spectrum of LGP climate states and including 20 mean sea-ice edges might distract from the primary message of the plot.

- Data availability: The DOI points to a lower-resolution (5cm) version of the dataset. As such the study is, therefore, not (yet) reproducible.

  - o The mm-scale data set is currently being uploaded to the Arctic Data Center

**References**

[1] Malmierca-Vallet, I., Sime, L. C., and the D–O community members: Dansgaard–Oeschger events in climate models: review and baseline Marine Isotope Stage 3 (MIS3) protocol, Clim. Past, 19, 915–942, https://doi.org/10.5194/cp-19-915-2023, 2023.
[2] Serreze, M. Rethinking the sea-ice tipping point. Nature 471, 47–48 (2011). https://doi.org/10.1038/471047a

[3] Livina, V. N. and Lenton, T. M.: A recent tipping point in the Arctic sea-ice cover: abrupt and persistent increase in the seasonal cycle since 2007, The Cryosphere, 7, 275–286, https://doi.org/10.5194/tc-7-275-2013, 2013.

[4] Rhines, Andrew, and Peter J. Huybers. "Sea ice and dynamical controls on preindustrial and last glacial maximum accumulation in central Greenland." Journal of Climate 27.23 (2014): 8902-8917.

**Citation**: https://doi.org/10.5194/egusphere-2024-1003-RC1

---

## Author Response (AR1)

**Reviewer 1 Summary**

An ultra-high resolution time series for central/eastern Greenland deuterium variability is presented. After correction for diffusion, 7-15-year variability is considered as represented and and interpreted in terms of temperature variability that changes between Greenland stadials and interstadials. Temperature change at the ice core site on decadal scales is compared to sea-ice and sea surface-temperature change in the North Atlantic, and interpreted as primarily arising from sea-ice variability.

Crucially, the authors suggest that a reduction of decadal temperature variability at the ice core site occurs centuries prior to the interstadial warming, and corroborate this with the analysis of model simulations and one high-resolution marine core.

The study is well-written, and the topic is relevant to Climate of the Past. However, there are a few points were more detail, or more stringent acknowledgement of uncertainties, is important.

**Major Points**

- Please acknowledge the assumption of the validity of the temperature interpretation of dD at sub-decadal timescales -- as you show, sea-ice variability is highly correlated with temperature, but it is not the only driver.

   o We acknowledge uncertainties in translating the water isotope record directly to temperature, especially at decadal and interannual scales. It is also acknowledged that further isotope modeling studies are required, as the current understanding of water isotopes at high-frequency scales is limited. The following preexisting lines of text highlight this:

      ▪ Line 48: "Stable isotopes of hydrogen ($\delta$D) and oxygen ($\delta^{18}$O) in polar ice cores provide information about local temperature and atmospheric circulation (Dansgaard, 1964)"

      ▪ Line 312: "It is possible that greater excursions in sea-ice concentration were driven by the enhanced latitudinal range between maximum summer and winter ice extents (Sadatzki et al. 2020), which presumably changed on a year-to-year basis. These seasonal variations would impart volatility in high-frequency temperature fluctuations on annual, interannual, and decadal scales via ice-atmosphere feedbacks. By extension, Rayleigh distillation and the isotopic signature of precipitation at EGRIP would also be impacted. An additional contribution to enhanced isotopic variability during stadial phases may stem from altered source-to-sink pathways. With large seasonal swings in the capping and exposure of the ocean surface by sea ice, evaporative sources upstream of EGRIP would also be altered. As a consequence, variability in both evaporative source signatures and temperature gradients of moisture transport to Northeastern Greenland would increase. An isotope enabled GCM is required to test this hypothesis."

- o  Another concern is that the isotopic signal we document may also be related to non-climate noise at high-frequencies, which changes across climate states (glacial interstadials vs stadials). It has been proposed that shifting accumulation rates may cause such a systematic influence on non-climate noise, as colder periods are associated with lower accumulation and thus greater precipitation intermittency and stratigraphic noise. To refute this, the authors have also now included the coevolution of EGRIP accumulation with 7-15 diffusion corrected variability in Figure A8. It shows that declines in variability lead accumulation shifts by hundreds of years for most D-O events. This suggests local depositional effects during cold stadial phases cannot account for the early shifts in 7-15 year variability, relative to D-O warming.

- o  An additional concern is that our results could by influenced by changing thermal gradients in the firn column below the surface when abrupt D-O warming occurs. We postulate that this could affect grain metamorphosis and vapor transport in a way that would keep pore spaces open longer, enhance mixing and decrease variability in sections of ice that are older than the onset of abrupt warming. We have added a section of text in the discussion to account for this possibility:

    - ▪  Line 383: "There is an additional explanation of the lead-lag result, though it currently lacks strong evidence. Water-isotope diffusion is a property not exclusively set at the ice sheet surface, but one that reflects the duration of the firn densification process. If the surface climate changes, this impacts concurrent precipitation in addition to the firn column below via temperature gradients, which are important for grain metamorphosis and vapor movements via overburden pressure and barometric pumping. Changes in interannual-to-decadal variability in ice that is deeper (i.e. older) than abrupt D-O warmings may be driven by shifting thermal gradients that affect grain metamorphosis and vapor transport in such a way to keep pore pathways open longer, thereby enhancing gas mixing and ultimately reducing water isotope variability in older ice. Such a mechanism is not currently captured in accepted firn models (e.g., Johnsen et al, 2000)."

- •  Consider, in the Discussion, the robustness of the model-based process interpretation considering a usefulness of model intercomparison (e.g., [1]), in particular given the complexity of sea-ice models. Similarly, the marine record is a (hand-picked) example, there would be potential for targeted synthesis work here.

    - o  We agree that there is always the possibility for more synthesis in any paleoproxy study. We do, however, feel that it is beyond the scope of our study, which aims to show the currently undocumented interannual variability that is associated with DO events. Similarly, as the reviewer is no doubt aware, a full intercomparison of the all of the models that show some sort of DO-like variability is non trivial. We show how one set of model simulations can be used to contextualize our results but leave it to other authors to write the manuscript devoted to intercomparing all

of the available DO-like simulations. Our paper adds a further observational constraint that any model that aims to simulate DO like variability must be able to explain. We look forward to discussing with our modelling colleagues how our record and others do and don't agree with DO simulating models.

- The figures need to be reworked. Standard red/green looks grey to quite a few people (Listen, e.g. to these people describing their experience https://www.youtube.com/watch?v=FKSOe5NK_qQ and imagine how distinguishable colors are in most of your figures...)

    o Figures 1, 3 and 4 have been reworked to remove all standard red/green color combinations

**Minor Comments**

- p1l26/27 This sentence is ambiguously phrased. (Why) should there be a phase offset, and should it be distinguishable? I read this as: "Across stadial/interstadial transitions proxy evidence showed in-phase changes in mean temperature/dust/sea-salt concentration/accumulation rate".

    o The author is stating (not questioning) that proxy evidence (mean temp/dust/sea salt/accumulation) exhibits in-phase shifts across stadial-interstadial transitions consistent with the cited literature (Capron et al., 2021).

- p1l29/30 You write that high-frequency interannual variability surrounding "mean temperature change" has not been investigated -- please clarify that by mean change you mean centennial-to-millennial scales, and by high-frequency interannual variability. From a lower-resolution marine point of view both timescales are "just" variability.

    o This has been rephrased

- p2l71 "tipping-point sea-ice displacement" - The concept of a threshold, below which the ice edge becomes unstable, and fast/complete retreat of perennial sea-ice cover occurs is debated [2,3]. Compared to the ice sheet, sea-ice itself has little memory, but small changes in the ice edge may lead to large impact warming. Please rephrase.

    o The author has removed the phrase "tipping-point"

- p3l85 and following: Investigating leads and lags, as well as interannual to multicentennial variability across the LGP and for different timescales ("mean" vs. "variability") and attributing it primarily to local temperature change assumes that dD is faithfully representing local EGRIP site temperature. This is not explicitly mentioned, but is permeating the study, and should be acknowledged explicitly. Sea-ice variability, independently of temperature change at the EGRIP site, can induce d18O variability [4], and, as the authors themselves show with the model-based correlations these variables are colinear. Isotope-enabled simulations could allow, to some extent, to disentangle these relationships.

    o See comment under first bullet point of "Major Points". We do not intend to assume dD faithfully represents local EGRIP site temperature and present multiple arguments backed by their results on why the water isotope record at

interannual and decadal scales is largely uncertain at present. We suggest isotope enabled modeling in future work.

- p4l141 timestep of 50-200 years -- presumably these are the shifts for the moving windows? Unclear.

  o Yes, this has been clarified in the manuscript on Line 145: "We spectrally analyze the EGRIP δD record in 400-year windows with a timestep (i.e. stepwise shift between adjacent windows) of 50 to 200 years, depending on desired temporal resolution"

- p5l154 correct: preserved

  o This has been corrected

- p6l204: instead of "spectrum of change" suggest rephrasing

  o This has been rephrased. Line 209 now reads: "The freshwater hosing and pinned sea-ice simulations give a range of mean temperatures at the location of EGRIP that represent a spectrum of possible mean LGP climate states. By including a spectrum, we reduce the bias that could arise from defining a single stadial or interstadial state with only one forcing mechanism."

- p7l217 increased depletion

  o This has been corrected

- p8l251 clarify "mean" timescale (see above)

  o This has been addressed. Line 258 now reads: "Thus, there is clear evidence that 7-15 year variability for D-O Events 2, 3, 4, 5.2, 6, 7, 8, 10, 12 and 13 leads centennial-scale mean temperature change at the onset of GI phases.

- p8l255 and following: How long are the simulations? How are the degrees of freedom and a significance for the correlations calculated? Are these step-wise simulations, and is the mean change then subtracted prior to correlation? The magnitude of the correlation is surprisingly high.  To what extent are these correlations representative for other models (given the fairly simple sea-ice model in HadCM3)?

  o All simulations are run for 500 years, branched from a control simulation (Pre-industrial/LGM) that has been run for many 1000s of model years. Climatologies are computed from the last 100 years. All simulations are self contained, independent, simulations and branch from the same control simulation at the same time - they are not "step-wise" in the sense that the simulation with the sea ice edge at 55N feeds the simulation with the sea ice edge at 50N etc. In fig 5 (a) and (b) no means are subtracted as the correlation is between the variability (variance) in sea ice concentration and temperature. In figs 6(b) and (c) the abscissae do not have the control mean temperature removed: we wish to establish the relationship between cold and warm states and variability, therefore must include the overall colder conditions that arise from the LGM climate. The correlations in fig 5(a) are indeed high as in this figure we are correlating temperature variability with itself - over EGRIP this is by definition 1. This map gives a sense of the spatial

autocorrelation for 7-15 year variability over the North Atlantic. Unsurprisingly over Greenland this is high. The correlations with sea ice are maximum at around 0.6, so sea ice variability can explain ~30% of the variance at EGRIP, thus the majority of the variance arises from other sources. Since no other model has been run across such a range of different forcings, it is impossible to say to what extent the results arise from HadCM3's quirks. However, over the 7-15 year timescale the atmosphere has little memory so coherent changes are driven by the ocean/ice system. Therefore, the magnitude of the correlations that we see arise from the dynamics of the atmosphere. Thus the relatively simple sea ice model in HadCM3 is not likely to bias the results.

- p10l350 Arguably, this is a single core site for which a reduction of sea-ice occurs prior to Greenland isotopic/temperature change, and a single climate model. The correlation patterns of sea-ice variability with EGRIP temperature in other models would be interesting. What is the age model of MD95-2010 based on, and what is the corresponding age uncertainty? Hopefully (or perhaps, evidently, from the results) not tie-points to GICC05. Perhaps this is an age model issue?

  o Text regarding sediment core age model taken directly from Sadatzki et al., 2020:

    ▪ "Accordingly, the age model of MD99-2284 is based on alignment of near-surface temperature signals and D-O climate transitions, independently verified and constrained by four distinct cryptotephra layers that were identified before and after the GS6GI5 transition as well as during GI6 and GI8 in both core MD992284 and the NGRIP (North Greenland Ice Core Project) ice core with a consistent geochemistry (33) (SI Appendix, Materials and Methods and Fig. S2). Moreover, the glacial sediment sections in both cores MD95-2010 and MD99-2284 reveal a very consistent variability in anhysteretic remanent magnetization (ARM), reflecting deep ocean circulation changes in the Nordic Seas (13, 34, 35), which closely resemble the D-O climate fluctuations recorded by the $\delta18O$ of the NGRIP ice core. This enables development of an age model for core MD95-2010 by stratigraphic alignment of its ARM record to that of MD99-2284 and the $\delta18O$ of the NGRIP ice core (SI Appendix, Materials and Methods and Fig. S2). Thereby, our sedimentary sea ice records are placed on the Greenland ice core chronology GICC05 (36) and can thus be directly compared with the RECAP sea ice record, which also has been transferred to the GICC05 chronology by alignment of the RECAP dust record to NGRIP $\delta18O$".

  o The above text has now been summarized in the manuscript on Line 363: "In this study, sediment core age models are tied to GICC05 using stratigraphic alignment of ARM (anhysteretic remanent magnetization), near-surface temperature, and cryptotephra layers with NGRIP $\delta^{18}O$."

  o We acknowledge further isotope-enabled modeling and high-resolution sea ice reconstructions are critical (Line 424): "Additionally, future isotope-enabled GCM studies may benefit from utilizing the high-frequency EGRIP variability timeseries, presented here, to constrain boundary conditions or benchmark model

output. We suggest targeted tests aimed at temporally reconciling the centennial-scale offset with sea-ice behavior to better understand regional North Atlantic climate change within the context of abrupt D-O warming. Analysis of high-resolution sediment proxy records from critical locations identified in this study may also clarify uncertainties."

- Fig. 3, 4, 6, 7 please avoid red/orange and green as dominant colors in figures (not colorblind friendly)

  o This has been addressed

- Fig. 5, perhaps add the mean position of the sea-ice edge in these figures for the LGM to aid interpretation.

  o We do not think this information would significantly aid interpretation as ice extent can be inferred from mean temperature in Figures A6 and A7, which are used to create the plots in Figure 5. Further, we use ~20 simulations over a spectrum of LGP climate states and including 20 mean sea-ice edges might distract from the primary message of the plot.

- Data availability: The DOI points to a lower-resolution (5cm) version of the dataset. As such the study is, therefore, not (yet) reproducible.

  o The mm-scale data set has been uploaded to the Arctic Data Center and the DOI has been changed accordingly

**Reviewer 2 Summary**

The study by Brashear et al. shows how stable water isotope interannual variability on the Greenland ice sheet changes throughout the Last Glacial, being stronger during stadials than interstadials, with peaks preceding D-O events by hundreds of years. They used CFA to measure high-resolution isotope data, spectral estimates for the correction of isotopic diffusion, and for estimating isotopic variance at interannual frequencies. They hypothesize that sea ice variability in the North Atlantic area and the mean temperature on the Greenland plateau are closely related to isotopic variability at the ice core location, underpinning this hypothesis by using HadCM3 models, and comparing ice sheet temperatures to sea ice dynamics. The study is important for advancing our understanding of the climate system, specifically Greenland variability and the North Atlantic Ocean and AMOC in relation to the global mean climate state, as well as the characteristics of abrupt climate changes by assessing sudden shifts of D-O events. They use adequate methods and contextualize their results within previous studies and hypotheses. We believe this paper includes interesting and relevant results suitable for publication in "Climate of the Past." However, we have some major concerns that should be addressed before publication, as well as some minor suggestions.

**Major concerns**:

- **Contribution of non-climatic noise on the changes of isotope variability:** The authors cautiously interpret isotopes and do not directly translate them to temperature, which aligns well with current knowledge of the uncertainties regarding isotope interpretation and isotope-temperature translations. They discuss altered source-sink pathways and evaporation sources upstream as other possible influences on isotope variability. Based on their thorough analysis of different frequencies (Figures A2, A5), their results should not be sensitive to time uncertainties within this (not layer counted) record.. As the analyzed core is highly influenced by ice flow, the authors could additionally state why they think upstream effects do not influence their results.
  Despite considering all these effects, the fast variability interpreted in this manuscript will still be influenced by non-climate noise. Even nearby ice core isotope records are found to be quite distinct from each other, especially at high frequencies (Münch & Laepple 2018). Estimates of the Signal to Noise Ratio (SNR) in the Greenland NGT stack (Hörhold et al. Extended Data Fig. 1b) suggest an SNR of 3-5 in this frequency band for a stack of 12 records; resulting in an SNR of around 0.3-0.4 for a single record such as EastGRIP. This in turn shows that the majority of the interpreted variance will likely be due to local depositional effects. Such noise components likely differ across climate states (e.g., GI vs. GS) and introduce isotope variability changes unrelated to climate. One characteristic that could hint towards such systematic influence could be a change in accumulation rates, which is strongly reduced in the cold phases compared to the Holocene. Lower accumulation rates in the last glacial coincide with more precipitation intermittency and stratigraphic noise. We, therefore, suggest that the authors show the accumulation history of the record, how it coevolves with the variability changes, and

discuss the possibility of state-dependent noise influencing the discovered changes in variability.

- o Ice flow at the EGRIP location causes the glacial portion of the water isotope record to originate ~200 km upstream near the ice divide (Gerber et al., 2021). A consequence of this may be thinning of interannual to decadal layers which could affect variability interpretations. Currently, we do not have a method to quantify this. One future possibility is to compare our results with those of a mm-scale water isotope record that is retrieved near the ice divide and not subject to ice flow/upstream effects. Still, we do not think the potential affect would change the primary message of our paper.
- o Accumulation rates in Greenland have been shown to move in phase with water isotopes during the LGP (Capron et al., 2021; Guillevic et al., 2013) where warm interstadials are associated with greater accumulation, and vice versa. We show in Figure A8 how EGRIP accumulation rates (Gerber et al., 2021) coevolve with high-resolution (i.e. 50 year timestep) 7-15 year variability. As expected, declines in variability lead accumulation shifts by hundreds of years for most D-O events. This suggests precipitation intermittency and stratigraphic noise during cold stadial phases cannot account for the early shifts in 7-15 year variability, relative to D-O warming. Additionally, we show in Figure A3 that diffusion length (which is significantly influenced by accumulation), also moves in sync with the water isotope record. Specifically, diffusion lengths in the time domain are lower during warm interstadials when accumulation is enhanced. Effects of accumulation on diffusion or the diffusion correction therefore also cannot account for the centennial-scale lead lag that we document. This further shows a robust result which is not overwhelming influenced by local depositional effects.
- o An additional concern is that our results could by influenced by changing thermal gradients in the firn column below the surface when abrupt D-O warming occurs. We postulate that this could affect grain metamorphosis and vapor transport in a way that would keep pore spaces open longer, enhance mixing and decrease variability in sections of ice that are older than the onset of abrupt warming. We have added a section of text in the discussion to account for this possibility:

    - ▪ Line 383: "There is an additional explanation of the lead-lag result, though it currently lacks strong evidence. Water-isotope diffusion is a property not exclusively set at the ice sheet surface, but one that reflects the duration of the firn densification process. If the surface climate changes, this impacts concurrent precipitation in addition to the firn column below via temperature gradients, which are important for grain metamorphosis and vapor movements via overburden pressure and barometric pumping. Changes in interannual-to-decadal variability in ice that is deeper (i.e. older) than abrupt D-O warmings may be driven by shifting thermal gradients that affect grain metamorphosis and vapor transport in such a way to keep pore pathways open longer, thereby enhancing gas mixing and ultimately reducing water isotope variability in older ice. Such a mechanism is not currently captured in accepted firn models (e.g., Johnsen et al, 2000)."

    - o

- o Hörhold et al., (2023) shows SNR for a stack of 12 records in the last 1000 years of the Holocene. Our record includes glacial stadials and interstadials, wherein the Hörhold analysis likely does not hold. Additional high-resolution analysis of deep ice cores would provide proof that the signal is repeatable across regions of Greenland, eliminating the concern that noise is causing much of the signal.

- **Uncertainty of the diffusion correction:** The authors estimate the diffusion length in the spectral domain. As shown by Jones et al., 2017 and by Kahle et al. 2018, in CFA systems, some noise is added to the isotopic signal on the preparation side of the system that, after the smoothing of the CFA system, leads to red noise at the higher frequency end (which would, with discrete measurements, be white), as visible in Appendix Figure A4. This red noise can interfere with the diffusion length estimate as it is difficult to distinguish from a diffused signal. Therefore techniques to account for this have been developed (Kahle et al.,  2018, Improved methodologies for continuous-flow analysis of stable water isotopes in ice cores, most authors from this paper are also on this new manuscript). The red noise might also influence the analyzed frequencies and the diffusion correction possibly amplifies this high frequency noise.
  We suggest that the authors use or at least discuss the diffusion length estimation method they introduced in Kahle et al., 2018  for CFA measured data. Further, the authors could show that their results are robust by elaborating on how the variability changes are also detectable on the diffused record (Figure 1c).
  - o Using the methods presented in this paper (Jones et al., 2018), we apply a correction to the diffused interval, which generally exists between the 2-20 year band. In the EGRIP record, analytical/red noise exists at higher frequencies (<1 year) which is orders of magnitude lower in its power density and therefore does not significantly influence the diffusion correction or subsequent calculations of high-frequency variability. We also use equally-spaced logarithmic bins to avoid weighing the correction towards higher frequencies which are more likely to be affected by analytical noise. This can be seen in Figure A4.
  - o The Kahle et al. (2018) and Jones et al., (2018) methods result in very similar estimations for diffusion length. Even if small variations exist, this is a moot point because the raw signal and the corrected signal in the 7-15 year band yield the same temporal evolution, just at varying magnitudes. In other words, the lead-lag relationship we document is a robust feature of the climate system and unrelated to the diffusion correction. The manuscript has been edited to make the above point clear on:
    - ▪ Line 232: "Another important detail is that the raw (i.e. non-diffusion corrected) 7-15 year variability record exhibits lower amplitudes, yet simultaneous shifts with the corrected record (Fig 1c). This demonstrates the ability of our correction to target signal attenuation by diffusion without incorporating uncertainty into the temporal evolution of the record. In other words, the signal we document is a robust feature of the climate system and not an artifact of the diffusion correction or laboratory analysis."

- **Variability leading abrupt change or variability just depending on the mean state?** We suggest that the authors interpret their results on the variability 'leading' abrupt climate change with more caution. They write, 'Such a large phase offset between two climate parameters in a Greenland ice core has never been documented for D-O cycles' (Lines 34ff). To play devil's advocate, at least visually, the minima in δD also seem to lead the onset of the interstadial periods (their Fig. 3). This may be due to the definition of the onsets, which is set at a certain magnitude of change in the proxy records over time (Rasmussen et al. 2014), combined with the typical shape of the isotope changes. The counter-hypothesis would thus be that the variability depends on the mean isotope value (their Figure 1b), and this dependency (which is interesting in itself) already explains the time-lag. We therefore suggest that the authors either refute this simpler counter-hypothesis, or if this is not possible, one down their interpretation of their results
  - To refute the counter-hypothesis, Figure 1b demonstrates a general relationship between 7-15 variability and mean Greenland climate (e.g. Holocene, Interstadial, Stadial), though there is significant overlap in the data for Interstadials and Stadials. A closer look at Figure 3 shows a substantial centennial-scale offset between shifts in variability and the following onset of abrupt warming, which partly explains the overlap. Though δD may reach an absolute minimum earlier in the stadial phase, the overwhelming interest in D-O Events is the rate and magnitude of change that occurs at a GI-GS transition which is well studied and defined according to Rasmussen et al., 2014. Between Figures 3 and 4, we show that 50% or more of the abrupt change in 7-15 year variability occurs prior to the onset of D-O Events (Line 254), indicating a decoupling between the two variables. The authors feel it is reasonable to state "Such a large phase offset between two climate parameters in a Greenland ice core has never been documented for D-O cycles"
  - We also state that there are large swings in variability between 27-14 ka b2k when D-O cycling is infrequent, further strengthening the argument that Greenland isotope mean and variability are decoupled (for reasons that must be further researched to fully understand).
    - Line 245: "It is important to note that we document multi-millennial excursions in variability occurring between 27-15 ka b2k wherein cold GS conditions persist uninterrupted by abrupt warming, with the exception of D-O Event 2 around 23 ka b2k. The excursions are comparable, yet generally smaller in magnitude than those occurring between 50-27 ka b2k when D-O cycling is relatively consistent."
    - Line 375: "Lastly, an inexplicable component of this study is the continuation of large excursions in high-frequency isotopic variability even when D-O cycling is turned off for long stretches (i.e. 27-15 ka b2k). In some cases, the fluctuations are comparable in magnitude to those occurring across prior GS-GI transitions. It seems a climate variability oscillation is inherent to the LGP background state, yet does not result in abrupt mean climate change (i.e. D-O Events) based on certain boundary conditions which remain to be seen. Due to the simultaneous occurrence

of the Last Glacial Maximum during this timeframe, an obvious factor to test in future studies is the height and extent of the Laurentide and Scandinavian Ice Sheets and their effects on climate variability."

**Minor comments**:

- 42 "Thus, both paleoclimate proxy evidence and model simulations suggest that sea ice plays a substantial role in high-frequency climate variability prior to D-O warming." - Argument unclear. You mean paleoclimate proxy evidence including the ice core records as well as the open ocean biomarkers? The ice cores alone do not evidence that, so maybe mention the biomarkers as being part of the "paleoclimate proxy evidence" you are referring to, or delete "Thus" as: "Both paleoclimate proxy evidence as well as these model simulations suggest…"
  - "Thus" has been removed and Line 43 has been rephrased to: "Together, paleoclimate proxy evidence and model simulations suggest that sea ice plays a substantial role in high-frequency climate variability prior to D-O warming."
- 63 References unclear. Which literature explains D-O warming events being related to sea ice and which literature just generally associates sea ice with abrupt warming? Do all of the studies do both? Then maybe add a : between the two sentences?
  - All references provide evidence of DO events being related to sea ice behavior
- 134 "On average, temporal differences in adjacent data points range from sub-weekly in the Holocene to sub-monthly during the LGP". Can you clarify what you mean by "adjacent"? Temporally closest together?
  - "Adjacent" meaning data points which are next to one another; this has been clarified on Line 139
- 151 pore "close-off"
  - This has been corrected
- 139 Can you state why you choose not to include lower frequencies - e.g., because of prior expectations regarding sea ice variability?
  - High-frequency variability is stated as the focus of our study beginning on Line 92: "Though studies assessing decadal-scale variability during the LGP exist (Boers et al., 2018; Ditlevsen et al., 2002), the data sets used were discretely sampled at cm-scale resolutions which may diminish or conceal important high-frequency climatic information (Fig. A1). Developments in continuous-flow sampling techniques have recently allowed for high-frequency analysis of water isotope variability in Antarctic ice cores during the LGP by preserving the amplitude of interannual-scale signals (Jones et al., 2017a; Jones et al., 2018). In the case of West Antarctica, a shift in LGP interannual isotopic variability was linked to broad changes in Pacific Basin teleconnection strength driven by reductions in Laurentide Ice Sheet topography and changing albedo (Jones et al., 2018). This study demonstrated that the drivers of high-frequency climate variability can temporally decouple from the drivers of mean local climate (e.g. temperature, accumulation, etc.), providing new insights about paleoclimate dynamics. In the northern high latitudes, sea ice varies substantially on multi-year and multi-decade bases, imparting variability into the climate system on similar timescales. The Greenland water isotope variability record may therefore provide

clues about high-frequency sea-ice variations as such shifts would affect the isotopic signature of precipitation via influences on both moisture source and atmospheric circulation."

- 154: persevered or better "preserved"?
  - This has been corrected
- 185: The method description is too short to be reproducible. If I understand it right, it needs to assume / assumes that 1.) $P_0(f)$ is not frequency dependent and the fit takes only place on frequencies lower than a manually chosen fc to ensure that the spectrum is dominated by the diffusion signal in this range of frequencies and measurement noise can be ignored.
  - The authors feel the description accurately and succinctly describes the methods used in this study and is consistent with prior studies (Jones et al., 2017b; Jones et al., 2018; Jones et al., 2023; Kahle et al., 2021)
    - $P_0(f)$ is still frequency dependent based on its definition in equation 3
    - It is unclear what the variable 'fc' is in reviewer comment, but it is correct that the correction fit is placed on frequencies affected by diffusion and not analytical noise
- Event 2.2 is mentioned for the first time, please define what that is. It's not in the table.
  - The following text has been added to table caption: "Due to the brief duration of G-I 2.1 and 2.2 (collectively referred to as D-O 2 in Table 1), one 400-year window is placed at the onset of G-I 2.2 (i.e. 23.34 ka b2k), which encompasses the entirety of G-I 2.2, a majority of G-I 2.1, and the short-lived stadial phase between each interstadial"
- F2: Please define 2.1 and 2.2 events
  - This has been addressed in captions of Figure 2 and Table 1
- F2 and F3: Could you please put the names/numbers of the D-O events into the graphic to make it easier to follow? Right now, if you write about a specific event, one has to check the table for the D-O event's time and then search for it in the graph.
  - This has been addressed on Figures 2 and 3

---

## Referee Report (RR1)

The study by Brashear et al. shows how stable water isotope interannual variability on the Greenland ice sheet changes throughout the Last Glacial, being stronger during stadials than interstadials, with peaks preceding D-O events by hundreds of years.

They adjusted minor irregularities in the text and further
· show the robustness of their results towards diffusion correction,
· discuss possible uncertainties of the diffusion estimates related to systematic density changes,
· show in a graphic how accumulation rates coevolve with the high frequency isotope variability.
With this, they address some of the issues we raised in the first review. We therefore think that the paper is in a good state for publishing.

We have just a few comments left to revise:

Two minor comments:

L.100: "*decouple from the drivers of mean local climate (e.g. temperature, accumulation, etc.)*" – use either "e.g.", or "etc." - Also, how does this study show that the drivers of high frequency isotope variability are temporally decoupled from accumulation? It does not, as it does not include any accumulation rate analysis.

Fig. 8A; Thanks for adding this graphic. In the review answer you state: "As expected, declines in variability lead accumulation shifts by hundreds of years for most D-O events. This suggests precipitation intermittency and stratigraphic noise during cold stadial phases cannot account for the early shifts in 7-15 year variability, relative to D-O warming". As changes in stratigraphic noise and precipitation intermittency with time cannot be quantified, while higher accumulation rate changes might facilitate signal preservation, (which do seems strongly correlated to high frequency variability), noise changes could still be a reason for variability changes, which should still be stated in the text as one (counter?) hypothesis.

And our comment on the method description was not solved and still needs to be adressed

**Review Round 1:**
- **185: The method description is too short to be reproducible. If I understand it right, it needs to assume / assumes that 1.) P0(f) is not frequency dependent and the fit takes only place on frequencies lower than a manually chosen fc to ensure that the spectrum is dominated by the diffusion signal in this range of frequencies and measurement noise can be ignored.**
  o The authors feel the description accurately and succinctly describes the methods used in this study and is consistent with prior studies (Jones et al., 2017b; Jones et al., 2018; Jones et al., 2023; Kahle et al., 2021)
  § Po(f) is still frequency dependent based on its definition in equation 3

**§ It is unclear what the variable 'fc' is in reviewer comment, but it is correct that the correction fit is placed on frequencies affected by diffusion and not analytical noise**

While we acknowledge that the specifics of the diffusion correction do not alter the results, we insist that the method description must be comprehensive enough to ensure full reproducibility, as this is standard good scientific practice.

Currently, it is unclear in which frequency range the fit is performed or how this range is determined (e.g., manually for each depth or using a single range). Additionally, it is unclear whether the fit is applied to $P$ or $\ln(P)$, as suggested in line 190: "Diffusion length, $\sigma_a$, can also be quantified as the slope, $m$, of a linear regression, $y$, fitted to $\ln[P(f)]$ versus $f^2$ of the diffused interval."

The references cited use different approaches. If I read it right, Kahle (2021) accounts for the red CFA spectra by fitting two Gaussian distributions, whereas Jones et al. (2018) uses only one Gaussian "fits to the frequency at which there is a distinct slope break in $\ln(PD)$." Therefore, simply citing these references does not provide the reader with a reproducible method.

---

## Author Response (AR2)

**Reviewer 2 Summary**

The study by Brashear et al. shows how stable water isotope interannual variability on the Greenland ice sheet changes throughout the Last Glacial, being stronger during stadials than interstadials, with peaks preceding D-O events by hundreds of years.

They adjusted minor irregularities in the text and further
· show the robustness of their results towards diffusion correction,
· discuss possible uncertainties of the diffusion estimates related to systematic density changes,
· show in a graphic how accumulation rates coevolve with the high frequency isotope variability.

With this, they address some of the issues we raised in the first review. We therefore think that the paper is in a good state for publishing.

We have just a few comments left to revise:

Two minor comments:

1) Fig. 8A; Thanks for adding this graphic. In the review answer you state: "As expected, declines in variability lead accumulation shifts by hundreds of years for most D-O events. This suggests precipitation intermittency and stratigraphic noise during cold stadial phases cannot account for the early shifts in 7-15 year variability, relative to D-O warming". As changes in stratigraphic noise and precipitation intermittency with time cannot be quantified, while higher accumulation rate changes might facilitate signal preservation, (which do seems strongly correlated to high frequency variability), noise changes could still be a reason for variability changes, which should still be stated in the text as one (counter?) hypothesis.
   - Its important to note that higher accumulation rates (interstadials) are associated with lower high-frequency variability in this study, which is in contradiction to your comment above
   - The following text has been included at line 385 to consider the effects of stratigraphic noise: " There are additional explanations of the lead-lag result, though they currently lack strong evidence. The first variable to consider is stratigraphic noise, which is non-climatic variability imparted to the water isotope record due to processes like precipitation intermittency, surface sublimation, and wind-driven snow erosion. Stratigraphic noise hinders the extraction of climate-induced high-frequency signals during low accumulation phases (e.g. LGP stadials), thus raising concerns that local depositional processes may also drive the results presented in this study. Unfortunately, the temporal evolution of stratigraphic noise cannot be quantified directly and currently, there are no LGP signal-to-noise ratio comparisons with nearby Greenland ice cores. Still, contributions of non-climatic noise are likely state dependent (e.g. GI vs GS

phases) and inherently linked to accumulation rate. EGRIP 7-15 year variability also exhibits a centennial lead-lag with accumulation, suggesting the primary driver for this deviation lies elsewhere (Fig. A8)."

2) While we acknowledge that the specifics of the diffusion correction do not alter the results, we insist that the method description must be comprehensive enough to ensure full reproducibility, as this is standard good scientific practice.

- Currently, it is unclear in which frequency range the fit is performed or how this range is determined (e.g., manually for each depth or using a single range).
    - The following text has been added on line 173 to address this comment: "The EGRIP $\delta$D diffused interval ranges between the 2-25 year band, and the Gaussian fit is optimized by manually adjusting this interval for each age window. This ensures the diffusion length is not calculated based on measurement noise, which occurs at an even higher frequencies and can be identified as a sudden bend in the slope of P(f)."
- Additionally, it is unclear whether the fit is applied to P or ln(P) , as suggested in line 190: "Diffusion length, sigma_a , can also be quantified as the slope, m , of a linear regression, y , fitted to ln[P(f)] versus f^2 of the diffused interval." The references cited use different approaches. If I read it right, Kahle (2021) accounts for the red CFA spectra by fitting two Gaussian distributions, whereas Jones et al. (2018) uses only one Gaussian "fits to the frequency at which there is a distinct slope break in ln(PD) ." Therefore, simply citing these references does not provide the reader with a reproducible method.
    - At line 190, we have clarified that this is an alternative approach to calculate diffusion length, in which the slope of a linear regression fitted to ln[P(f)] vs f^2 represents the diffusion length. The citations associated with this section are (Jones et al. 2017b, 2018): "Diffusion length, $\sigma_a$, can be alternatively quantified as the slope, *m*, of a linear regression, *y*, fitted to ln[P(f)] versus f$^2$ of the diffused interval. Uncertainty bounds are defined as maximum and minimum slopes within one standard deviation of *y* (Jones et al. 2017b, 2018) (Fig. A4)."
    - Kahle (2021), along with several other papers, is cited earlier in the methods section as an example of a study that uses spectral analysis to estimate cumulative mean water isotope diffusion in a water isotope record. Kahle (2021) is not meant to represent the methods outlined. Rather, we cite Jones et al. (2017b, 2018) throughout the description of methods used where only one Gaussian fit is used.

---

## Author Response (AR3)

l.388: precipitation intermittency is not part of the stratigraphic noise (also it was not mentionned as such by reviewer 2). So please modify and invoke precipitation intermittency and/or stratigraphic noise

- Comment from reviewer 2 in round 1 revisions: "Lower accumulation rates in the last glacial coincide with more precipitation intermittency and stratigraphic noise. We, therefore, suggest that the authors show the accumulation history of the record, how it coevolves with the variability changes, and discuss the possibility of state-dependent noise influencing the discovered changes in variability."
- The following citations have been added to line 386 to invoke precipitation intermittency, sublimation and snow erosion as variables contributing to stratigraphic noise: The first variable to consider is stratigraphic noise, which is non-climatic variability imparted to the water isotope record due to processes like precipitation intermittency, surface sublimation, and wind-driven snow erosion (Fisher et al., 1985; Helsen et al., 2005; Town et al., 2008; Zuhr et al., 2021).

l. 421: This paragraph appears a bit weird at the end. Perhaps, it would be better to introduce it a bit because it is the conclusion of the discussion or perhaps a note to the reader. At least, add something like "Finally" or "As a final note" ?

- Line 421 now reads: "As a final note, evidence of centennial-scale lead times in sea-ice proxies from Norwegian sediment cores as compared to Greenland abrupt warming, as well as the model evidence presented here, suggest the weight of evidence is currently in favor of a regional climate driver rather than firn dynamics or stratigraphic noise."